# CONTINUAL LLAVA: CONTINUAL INSTRUCTION TUNING IN LARGE VISION-LANGUAGE MODELS

## ABSTRACT

Instruction tuning constitutes a prevalent technique for tailoring Large Vision Language Models (LVLMs) to meet individual task requirements. To date, most of the existing approaches are confined to *single*-task adaptation, whereas the requirements in real-world scenarios are inherently varied and continually evolving. Thus an ideal LVLM should sustain continual instruction tuning in the face of *stream*-task distributions (*i.e.*, different domains, emerging capabilities, and new datasets) while minimizing the forgetting of previously acquired knowledge. To achieve this, we propose a new benchmark for **CO**ntinu**A**l in**S**truction **T**uning on LVLMs (**COAST**), which encompasses the aforementioned domain-incremental, capability-incremental, and dataset-incremental configurations. In terms of methodology, we propose **Continual LLaVA**, a rehearsal-free method tailored for continual instruction tuning in LVLMs. To circumvent the additional overhead associated with experience replay, we freeze LVLMs and construct the *dual increment embeddings* for each input instruction to facilitate parameter-efficient tuning. Specifically, the increment embeddings can be decomposed into two principal components: 1) *intrinsic* increment embeddings to encode task-specific characteristics. To achieve this, we set up a low-rank pool containing candidate embeddings, from which we select the relevant ones based on their similarity with the user instructions; 2) *contextual* increment embeddings to investigate the inter-dependencies across tasks. In this regard, the low-rank embeddings chosen in the previous tasks are aggregated via learnable weighted sum to provide complementary hints. Extensive experiments indicate that the proposed Continual LLaVA outperforms previous methods by significantly reducing the forgetting during the continual instruction tuning process.

## 1 INTRODUCTION

Large Language Models (LLMs) such as GPT (Achiam et al., 2023; Brown et al., 2020) and LLaMA (Touvron et al., 2023a;b) have demonstrated impressive abilities in comprehending user instructions and generating reliable responses. Building upon these achievements, recent advancements in Large Vision-Language Models (LVLMs) (Li et al., 2023b; Alayrac et al., 2022; Zhu et al., 2023a; Liu et al., 2024b; Wu et al., 2023; Li et al., 2024a; Zhan et al., 2024) integrates visual perception capabilities into LLMs, which has sparked considerable research interest.

Beyond the language understanding and generation ability, one prominent characteristic of LLMs and LVLMs is the emergent capability of instruction following (Ouyang et al., 2022; Zhang et al., 2023b), *i.e.*, faithfully responding to specific instructions and adhering to human preference. Instruction tuning enables LVLMs to generalize to unseen tasks by following task-specific instructions. Currently, most existing LVLMs are finetuned on the *single* instruction-tuning dataset. However, users' requirements are constantly evolving in practical applications. The robust and flexible LVLMs are expected to be continuously fine-tuned with *stream* instruction tuning datasets without the "catastrophic forgetting" (McCloskey & Cohen, 1989) of previously learned knowledge.

Compared to the well-defined per-category continual learning in image classification or object detection (Wang et al., 2024), the continual instruction tuning setting in LVLMs has not been clearly established. To this end, we collect and re-purpose existing benchmarks to construct a novel benchmark for **CO**ntinu**A**l in**S**truction **T**uning (**COAST**) on LVLMs. Specifically, we set up three contin-

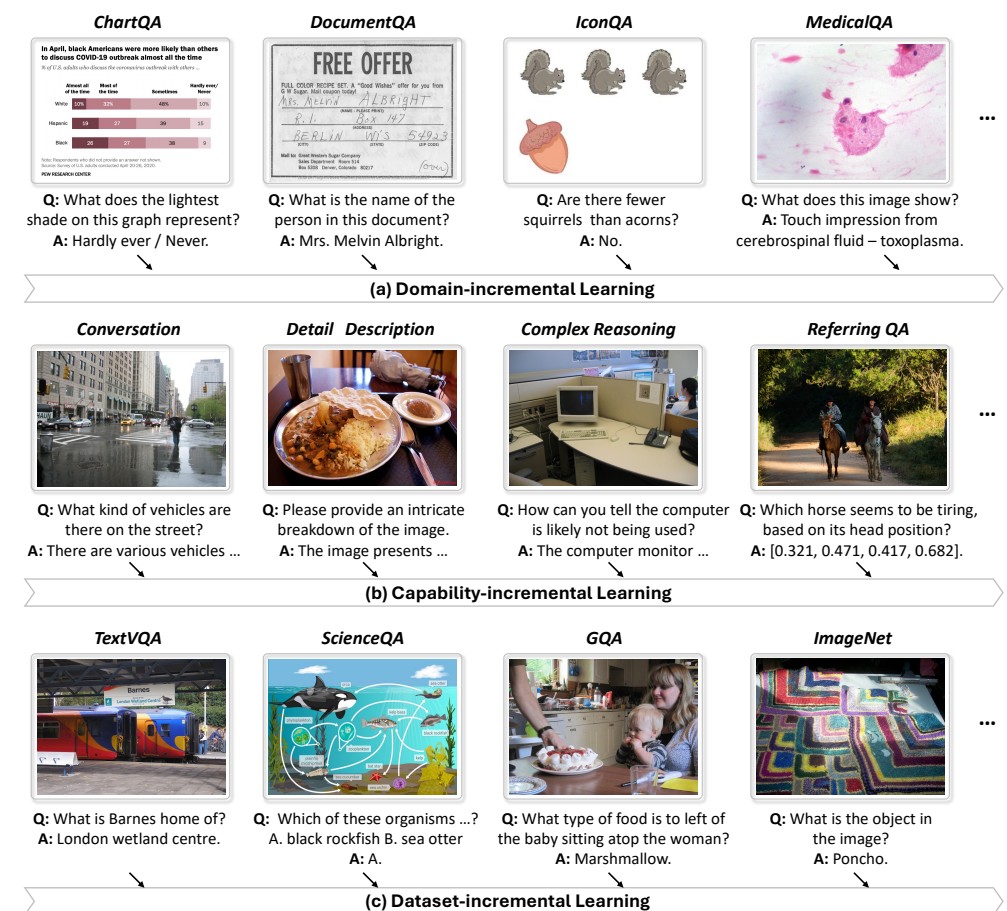

Figure 1: **COAST benchmark for continual instruction tuning** including (a) domain-incremental, (b) capability-incremental, and (c) dataset-incremental learning settings.

ual learning settings: **1) Domain-incremental**: As shown in Figure 1 (a), it aims to emulate the scenario where LVLMs are consistently adapted to different domains, *e.g.*, `chartqa`, `documentqa` and `iconqa`; **2) Capability-incremental**: This setting evaluates LVLMs' capacity to progressively acquire and integrate new functional capabilities, *e.g.*, `conversation`, `complex reasoning` and `detail description` in Figure 1 (b); **3) Dataset-incremental**: In this setting, LVLMs are exposed to cumulatively diverse datasets, assessing their ability to adapt and generalize across a range of dataset distributions (*c.f.* Figure 1 (c)). Based on the proposed COAST benchmark, we experiment and find that the intuitive sequential training of LVLMs, *i.e.*, training on new tasks[1] with initial weights from prior training, experiences significant performance degradation (*c.f.* Sec 4.2), which necessitates the development of a continuous instruction tuning method for LVLMs.

In this paper, we propose **Continual LLaVA**, a lifelong LVLM that continually adapts to new domains, learns new capabilities, or incorporates new datasets like humans. Inspired by the success of LoRA (Hu et al., 2021) in parameter-efficient tuning (Ding et al., 2023), we take one step further to construct a *low-rank pool*, which consists of a set of learnable *increment embeddings* generated by the low-rank decomposition. Different from the category-wise continual learning in image classification (Wang et al., 2022b), we construct the increment embeddings from two aspects: **1) Intrinsic Increments:** Each task has its distinct characteristic and necessitates unique increments for task-specific instruction tuning. For example in Figure 1 (a), LVLMs for `chartqa` typically require statistical and graphical literacy while LVLMs for `medicalqa` need domain knowledge of anatomy, physiology, and pathology. To achieve this, the corresponding increment embeddings are selected according to the similarity with user instruction and adapted into LVLMs while keeping the

---

[1]In this paper, we use the term "task" to collectively refer to domain, capability, or dataset.

pre-trained LVLM frozen; **2) Contextual Increments:** Each task exhibits correlations with other ones, indicating inter-dependencies that can be leveraged to enhance knowledge transfer and generalization across tasks. For example in `referring QA` of Figure 1 (b), when asked to find the coordinates of "the tired horse", LVLMs must *reason* about spatial relationships of the existing two horses to correctly identify the referred one. Thus we aggregate the selected increments in previous tasks via learnable weights to explicitly exploit the shared knowledge among different tasks[1].

In summary, our contributions are in three-folds:

- We collect and re-purpose existing benchmarks to curate COAST as a continual instruction tuning benchmark with the domain-wise, capability-wise and dataset-wise incremental learning settings.

- We propose a novel Continual LLaVA model, a lifelong LVLM to facilitate the continual instruction tuning across different domains, functional capabilities, or diverse datasets through learning parameterized intrinsic and contextual knowledge.

- Experimental results have manifested the state-of-the-art performance of our Continual LLaVA. For example on COAST-domain, Continua LLaVA surpasses the sequential training by achieving 13.06% absolute improvement in average accuracy and 13.25% reduction in average forgetting.

## 2 RELATED WORK

**Large Vision-Language Models.** LVLMs (Alayrac et al., 2022; Li et al., 2023b; Liu et al., 2024b; Sun et al., 2024; Jin et al., 2023) have garnered substantial research attention by advancing and integrating visual understanding and generation capabilities into LLMs (Achiam et al., 2023; Anil et al., 2023). A typical LVLM can be abstracted into three components, *i.e.*, a pre-trained vision encoder (Radford et al., 2021; Kirillov et al., 2023), a pre-trained LLM (Chiang et al., 2023a), and an interface connector in between. The primary attempt Flamingo (Alayrac et al., 2022) fuses the visual embedding into textual tokens of LLMs via cross-modal attention. The following works convert visual embeddings into LLM-understandable tokens using multi-layer perceptron (Liu et al., 2024b; Sun et al., 2024), Q-former (Bai et al., 2023; Li et al., 2023b), or discretization tokenizer (Jin et al., 2023). Our Continual LLaVA follows the LLaVA-styled (Liu et al., 2024b) multi-layer perceptron architecture due to its efficient setup, outstanding performance, and extensive usage.

**Instruction Tuning in LVLMs.** LVLMs typically undergo the following stages of training, *i.e.*, pre-training (Lin et al., 2024), instruction tuning (Ouyang et al., 2022), and optional alignment tuning (Sun et al., 2023; Ziegler et al., 2019). Among them, instruction tuning boosts the zero-shot or few-shot performance by generalizing LVLMs into unseen tasks by following task-specific instructions (Wei et al., 2022; Park et al., 2024). To achieve this, open-source LVLMs generate high-quality instruction-tuning datasets through self-instruction (Wang et al., 2023c), which prompts closed-source LLMs (Achiam et al., 2023) to generate instruction-following data using a few in-context examples. Cambrian (Tong et al., 2024) has compiled all the available datasets and restructured them into instruction tuning format. Most existing approaches limit their focus to instruction tuning for a specific task, overlooking the essential area of continuous instruction tuning for stream tasks.

We offer a detailed review of the limited research on continual learning for LVLMs, including recent pre-print works (Chen et al., 2024; Zhu et al., 2024; Zheng et al., 2024; He et al., 2023; Zhai et al., 2023). EMT (Zhai et al., 2023) focuses on the influence of fine-tuning LVLMs on image classification performance of the vision encoder, rather than on the instruction-following ability that our study prioritizes. While (Zhu et al., 2024) examines the performance trade-off between pre-trained and fine-tuned models, it does not involve the continual tuning in the more challenging streaming data. The pre-print works (Chen et al., 2024; Zheng et al., 2024; He et al., 2023) focus on continual instruction tuning but are limited to the dataset-incremental scenario. In contrast, we advance them by categorizing continual instruction tuning along three dimensions, (*i.e.*, domain, capability, and dataset), thoroughly addressing practical and real-world demands.

**Continual Learning.** Inspired by the incremental learning pattern (Chen & Liu, 2022; Wang et al., 2024) observed in human brains, continual learning focuses on the sequential training paradigm on a series of tasks with the expectation of maintaining performance across all tasks (Wang et al., 2024; Lee et al., 2017; McCloskey & Cohen, 1989). Early attempts adopt the regularization methodology (Kirkpatrick et al., 2017; Li & Hoiem, 2017; Feng et al., 2022; Yang et al., 2024) to penalize the updates to parameters that are important for previous tasks. Subsequent architecture-based works

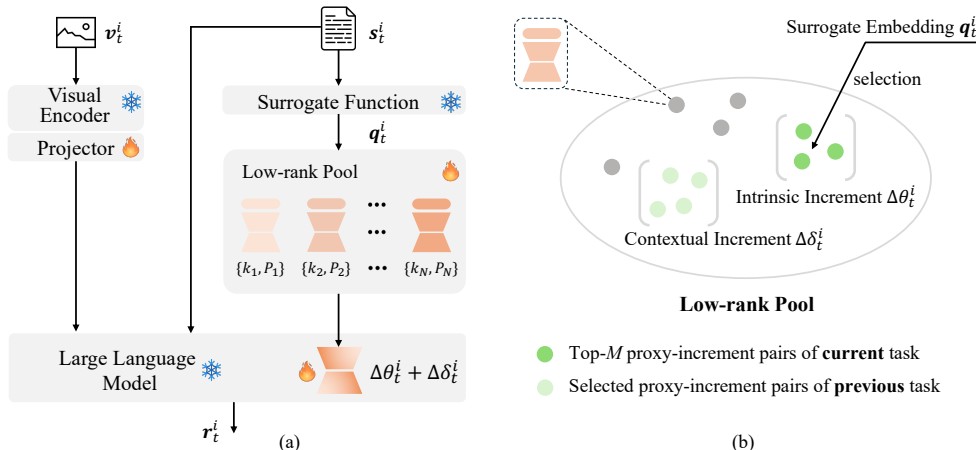

Figure 2: **(a) An overview of Continual LLaVA.** The $i$-th input image of $t$-th task $\boldsymbol{v}_t^i$ is processed via the pre-trained visual encoder followed by a linear projection layer. The corresponding textual instruction $\boldsymbol{s}_t^i$ is embedded as $\boldsymbol{q}_t^i$ by a frozen surrogate function. The low-rank pool contains $N$ learnable *proxy-increment* embedding pairs $\{\boldsymbol{k}_n, \boldsymbol{P}_n\}_{n=1}^N$, where the dual increment embeddings are selected according to the cosine similarity with $\boldsymbol{q}_t^i$. **(b) The schematic illustration of the dual increment embeddings.** We construct intrinsic embeddings $\Delta\theta_t^i$ by aggregating the top-$M$ items from the low-rank pool based on their similarity to $\boldsymbol{q}_t^i$. Contextual increments $\Delta\delta_t^i$ are generated by integrating the selected embeddings from all the previous tasks via learnable weights.

differentiate tasks via parameter isolation (Mallya & Lazebnik, 2018; Serra et al., 2018), dynamic architectures (Yoon et al., 2018; Hung et al., 2019), or modular networks (Shen et al., 2019). Another kind of rehearsal-based methods (Bonicelli et al., 2022; Chen & Chang, 2023; Lin et al., 2023) constructs the memory buffer to store and replay past data to prevent forgetting. To reduce buffer overhead, prompt-based methods (Wang et al., 2022b; Smith et al., 2023; Wang et al., 2022a; Li et al., 2024b) exploit learnable prompts to serve as the succinct episodic memory system for rehearsal-free continual learning. Different from the category-wise continual learning in image classification or object detection (Wang et al., 2024), this work demonstrates the potential of LVLMs to be continually adapted to novel tasks under the instruction tuning paradigm.

## 3 METHOD

The schematic illustration of Continual LLaVA is demonstrated in Figure 2. In Sec. 3.1, we present the overview of Continual LLaVA including visual & textual embeddings, dual increment embeddings, and LLM. Then we detail the proposed intrinsic and contextual increment embedding mining in Sec. 3.2. Finally, the adaption procedure and optimization objectives are presented in Sec. 3.3.

### 3.1 OVERVIEW

The proposed Continual LLaVA is trained with a chain of instruction-tuning tasks[1] at the domain, capability, or dataset levels. Suppose that we have the stream instruction-tuning tasks $\{\mathcal{D}_t\}_{t=1}^T$, where each task $\mathcal{D}_t = \{\boldsymbol{v}_t^i, \boldsymbol{s}_t^i, \boldsymbol{r}_t^i\}_{i=1}^{|\mathcal{D}_t|}$ comprises the triplet of the input image $\boldsymbol{v}_t^i$, instruction $\boldsymbol{s}_t^i$, and output response $\boldsymbol{r}_t^i$, $i = \{1, 2, \cdots, |\mathcal{D}_t|\}$.

Structurally, Continual LLaVA comprises the following four major components.

- *Visual Embedding*: Given the $i$-th input image of $t$-th task $\boldsymbol{v}_t^i$, we follow (Liu et al., 2024b;a) to extract the visual embeddings. Specifically, we use the pre-trained CLIP visual encoder ViT-L/14 (Radford et al., 2021) followed by a linear projector to convert the visual embeddings into LLM understandable space. In experiments, the CLIP encoder is kept frozen and the linear projector is initialized using the pre-trained weights from (Liu et al., 2024a).
- *Textual Embedding*: For the input instruction $\boldsymbol{s}_i^t$, we adopt the widely used BPE tokenizer (Sennrich et al., 2016) to obtain the textual embeddings.

- *Dual Increment Embedding*: We establish a dual increment embedding framework consisting of intrinsic and contextual increment embeddings to capture and encode both the inherent characteristics and the contextual information for each input instruction.

- *Large Language Model*: Finally, LLM takes the visual embeddings, textual embeddings, and dual increment embeddings as input and generates the desired responses. The vanilla weights of LLM are kept frozen and only the mined dual increment embeddings are updated. We select Vicuna (Chiang et al., 2023b) as LLM for our experimental studies.

## 3.2 DUAL INCREMENT EMBEDDING MINING

**Intrinsic Increment Embedding:** We set up a low-rank pool to serve as a flexible and dynamic memory enabling Continual LLaVA to retrieve relevant information. Specifically, the low-rank pool consists of $N$ learnable proxy-increment embedding pairs, *i.e.*, $\{\boldsymbol{k}_n, \boldsymbol{P}_n\}_{n=1}^N$. The proxy embeddings $\{\boldsymbol{k}_n\}_{n=1}^N$ are used for the embedding selection, while increment embeddings $\{\boldsymbol{P}_n\}_{n=1}^N$ are adapted into LVLMs for efficient tuning. $\boldsymbol{P}_n \in \mathbb{R}^{D \times D}$ is generated as the product of learnable matrices $\boldsymbol{A}_n \in \mathbb{R}^{D \times R}$ and $\boldsymbol{B}_n \in \mathbb{R}^{R \times D}$, $R \ll D$, to enforce low rank.

$$\boldsymbol{P}_n = \boldsymbol{A}_n \cdot \boldsymbol{B}_n. \tag{1}$$

The input instructions take on the responsibility of selecting the intrinsic increment embeddings from the low-rank pool. To achieve this, we firstly employ Sentence-BERT (Reimers & Gurevych, 2019) to encode $\boldsymbol{s}_t^i$ as the *surrogate* embedding $\boldsymbol{q}_t^i \in \mathbb{R}^{D \times 1}$, where $\boldsymbol{s}_t^i$ denotes the $i$-th instruction of $t$-th task. Then we compute the cosine similarity between the surrogate embedding $\boldsymbol{q}_t^i$ and all the proxy embeddings $\boldsymbol{k}_n$ within the pool, $n \in [1, N]$. The proxy embedding and corresponding increment embeddings with the top-$M$ similarity scores are selected as follows.

$$\mathcal{I} = \{i_1, i_2, \cdots, i_M\} = \arg \mathrm{top}_{n \in [1,N]} \cos\left(\boldsymbol{k}_n, \boldsymbol{q}_t^i\right), \tag{2}$$

where $\mathcal{I}$ is the selected index set and $\cos(\cdot, \cdot)$ represents the cosine similarity computation. Thus the selected proxy and increment embeddings are denoted as $\{\boldsymbol{k}_{i_m}\}_{m=1}^M$ and $\{\boldsymbol{P}_{i_m}\}_{m=1}^M$, respectively. Finally, the intrinsic increment embedding is generated as follows by aggregating the selected increment embeddings in a $\mathrm{softmax}$ manner.

$$\Delta\theta_t^i = \frac{\sum_{m=1}^M \cos\left(\boldsymbol{q}_t^i, \boldsymbol{k}_{i_m}\right) \cdot \boldsymbol{P}_{i_m}}{\sum_{m=1}^M \cos\left(\boldsymbol{q}_t^i, \boldsymbol{k}_{i_m}\right)}, \tag{3}$$

where $\Delta\theta_t^i$ is the intrinsic increment embedding for the $i$-th data instance of $t$-th task.

**Contextual Increment Embedding:** We construct the contextual increment embeddings by integrating the learned embeddings from the previous tasks to provide complementary task-wise correlations. To achieve this, we maintain a task-wise set $\mathcal{Z}_t, t \in [1, T]$, to record all the selected increment embeddings in each task via Eq. 2. For the $t$-th task, the contextual increments are generated in a weighted sum of $\mathcal{Z}_l$ covering all the previous tasks, $l \in [1, t]$.

$$\Delta\delta_t^i = \sum_{l=1}^t \boldsymbol{w}_l \, \mathrm{sg}(\overline{\mathcal{Z}}_l), \tag{4}$$

where $\Delta\delta_t^i$ represents the contextual increment embedding for the $t$-th task. $\boldsymbol{w}_l \in [0, 1]$ is the learnable weight. $\overline{\mathcal{Z}}_l$ denotes the instance-wise average pooling results of the set $\mathcal{Z}_l$. Note that we freeze the previously learned $\overline{\mathcal{Z}}_l$ via the stop-gradient function $\mathrm{sg}(\cdot)$, which behaves like the identity function during the forward pass, but has zero gradients when computing the backward pass.

## 3.3 ADAPTATION TO LVLMS

**Adaptation to LVLMs:** Following (Hu et al., 2021), we freeze all the pre-trained weights of LVLMs and only selectively add and update the mined intrinsic and contextual increment embeddings. Here naturally arises the question of where to insert the selected increment embeddings. Recall that there exist four linear projection layers within the multi-head attention computation (Devlin, 2018), *i.e.*, the `query`, `key`, `value`, and `output` projection (*c.f.* Figure 4 in Appendix). Our experiments in Sec.4.3 show that re-parameterizing all four linear projection layers is unnecessary and we choose

---

**Algorithm 1** The training pipeline of Continual LLaVA

---

**Input:** Stream data $\{\mathcal{D}_1, \ldots, \mathcal{D}_T\}$, $\mathcal{D}_t = \{(\boldsymbol{v}_t^i, \boldsymbol{s}_t^i, \boldsymbol{r}_t^i)\}_{i=1}^{|\mathcal{D}_t|}$, where $\boldsymbol{v}_t^i$, $\boldsymbol{s}_t^i$ and $\boldsymbol{r}_t^i$ denote $i$-th input image, instruction and response in $t$-th task, respectively.
**Learnable Parameters:** Proxy embeddings $\{\boldsymbol{k}_n\}_{n=1}^N$; Increment embeddings $\{\boldsymbol{P}_n\}_{n=1}^N$.
**Hyper-parameters:** Pool size $N$; Selected number $M$; Task number $T$; Learning rate $\eta$.

1: **for** $t = 1, \ldots, T$ **do**
2:     $\mathcal{Z}_t \leftarrow \emptyset$                      ▷ Initialize selected increment embedding set for $t$-th task
3:     **if** $t > 1$ **then**
4:         $\{\boldsymbol{w}_1, \boldsymbol{w}_2, \cdots, \boldsymbol{w}_t\} \leftarrow \text{Parameter}(t)$       ▷ Initialize learnable vector with length $t$
5:     **end if**
6:     **for** $(\boldsymbol{v}_t^i, \boldsymbol{s}_t^i, \boldsymbol{r}_t^i) \in \mathcal{D}_t$ **do**             ▷ Input image, instruction and response
7:         Extract surrogate embedding $\boldsymbol{q}_t^i = \text{Sentence-BERT}(\boldsymbol{s}_t^i)$
8:         Compute cosine similarities between $\boldsymbol{q}_t^i$ and proxy embeddings $\boldsymbol{k}_n$ as $\cos(\boldsymbol{q}_t^i, \boldsymbol{k}_n)$
9:         Obtain index set $\mathcal{I} = \{i_1, i_2, \cdots, i_M\}$ with top-$M$ highest similarities via Eq. 2
10:        $\mathcal{Z}_t \leftarrow \mathcal{Z}_t \cup \{\boldsymbol{P}_{i_m}\}_{m=1}^M$              ▷ Update selected embeddings
11:         Compute intrinsic increment embedding $\Delta\theta_t^i \leftarrow \frac{\sum_{m=1}^M \cos(\boldsymbol{q}_t^i, \boldsymbol{k}_{i_m}) \cdot \boldsymbol{P}_{i_m}}{\sum_{m=1}^M \cos(\boldsymbol{q}_t^i, \boldsymbol{k}_{i_m})}$
12:         **if** $t > 1$ **then**
13:             Compute contextual increment embedding $\Delta\delta_t^i \leftarrow \sum_{l=1}^t \boldsymbol{w}_l \, \text{sg}(\overline{\mathcal{Z}}_l)$
14:         **end if**
15:         Re-parameterize LLMs via Eq. 5
16:         Gradient back-propagation to update $\boldsymbol{k}_{i_m} \leftarrow \boldsymbol{k}_{i_m} - \eta\nabla_{\boldsymbol{k}_{i_m}}\cos(\boldsymbol{q}_t^i, \boldsymbol{k}_{i_m})$    ▷ *c.f.* Eq. 6
17:         Gradient back-propagation to update $\Delta\theta_t^i \leftarrow \Delta\theta_t^i + \eta\nabla_{\Delta\theta_t^i}\mathcal{L}_{\text{ar}}(\boldsymbol{r}_i^t; \Delta\theta_t^i, \Delta\delta_t^i)$
18:         Gradient back-propagation to update $\Delta\delta_t^i \leftarrow \Delta\delta_t^i + \eta\nabla_{\Delta\delta_t^i}\mathcal{L}_{\text{ar}}(\boldsymbol{r}_i^t; \Delta\theta_t^i, \Delta\delta_t^i)$
19:     **end for**
20: **end for**

---

only to adapt the `output` linear projection for cost savings. Considering a specific `output` linear layer with pre-trained weight matrix $\boldsymbol{W}_0 \in \mathbb{R}^{d \times d}$, it is updated as follows.

$$\boldsymbol{y} = \boldsymbol{W}'\boldsymbol{x} = (\boldsymbol{W}_0 + \Delta\theta_t^i + \Delta\delta_t^i)\boldsymbol{x}, \tag{5}$$

where $\boldsymbol{x}$ denotes the input feature and $\boldsymbol{y}$ is the corresponding output. $\boldsymbol{W}'$ represents the adapted weights. $\Delta\theta_t^i$ and $\Delta\delta_t^i$ are respectively generated by Eq. 3 and Eq. 4. The pre-trained weights $\boldsymbol{W}_0$ are kept frozen and only the increment embeddings $\Delta\theta_t^i$ and $\Delta\delta_t^i$ are optimized.

**Optimization:** As shown in Algorithm 1, the overall optimization undergoes a two-stage training, *i.e.*, the first stage for the alignment between surrogate embeddings and proxy embeddings while the second stage for LLM auto-regressive training. For the first stage, we optimize the selected proxy embeddings $\{\boldsymbol{k}_{i_m}\}_{m=1}^M$ by pushing them close to the frozen surrogate embedding $\boldsymbol{q}_t^i$.

$$\mathcal{L}_{\text{align}} = -\sum_{m=1}^M \cos(\boldsymbol{q}_t^i, \boldsymbol{k}_{i_m}). \tag{6}$$

For the second stage training of Continual LLaVA, we adopt the conventional auto-regressive loss $\mathcal{L}_{\text{ar}}(\boldsymbol{r}_i^t; \Delta\theta_t^i, \Delta\delta_t^i)$ with the parameterized increment embeddings $\Delta\theta_t^i$ and $\Delta\delta_t^i$, where $\boldsymbol{r}_i^t$ denotes the response of the $i$-th data instance of the $t$-th task.

## 4 EXPERIMENTS

### 4.1 EXPERIMENTAL SETUP

**COAST Benchmark Construction.** We set up the COAST benchmark for continual instruction tuning on LVLMs. COAST contains the domain-incremental, capability-incremental, and dataset-incremental settings. **1) COAST-domain**: We select four different domain tasks including ChartQA (Masry et al., 2022), DocVQA (Mathew et al., 2021), IconQA (Lu et al., 2021), and MedicalQA (He et al., 2020). We use the instruction-following format of these datasets curated by (Tong et al.,

Table 1: **Evaluation results (%) of continual instruction tuning on COAST-domain.** "Avg." and "Fgt." represent average accuracy and average forgetting, respectively. "Reh.", "Seq." and "Joint" denote rehearsal, sequential and joint training.

| Methods | #Params↓ | Avg.↑ | Fgt.↓ | ChartQA | DocVQA | IconQA | MedicalQA |
|---------|----------|-------|-------|---------|--------|--------|-----------|
| Joint | 6.76B | 42.79 | — | 21.99 | 20.08 | 64.37 | 64.73 |
| CODA | 0.75M | 36.06 | 2.72 | 15.03 | 16.93 | 58.96 | 53.33 |
| Dual | 0.75M | 35.80 | 2.79 | 14.92 | 16.77 | 58.60 | 52.92 |
| L2P | 0.75M | 35.06 | 2.91 | 14.77 | 16.73 | 57.55 | 51.20 |
| LWF | 6.76B | 27.06 | 15.05 | 14.07 | 13.19 | 37.93 | 43.05 |
| EWC | 6.76B | 25.82 | 15.23 | 13.73 | 11.89 | 35.12 | 42.53 |
| Reh. | 6.76B | 24.92 | 15.61 | 13.10 | 11.20 | 34.83 | 40.53 |
| Seq. | 6.76B | 24.02 | 15.83 | 11.77 | 11.29 | 33.73 | 39.27 |
| **Ours** | **0.75M** | **37.08** | **2.58** | **15.30** | **17.82** | **60.71** | **54.50** |

2024). To ensure balance between tasks, we sample the same 20,000 instances from each domain data for training and 5,000 instances for evaluation. **2) COAST-capability**: We specifically focus on the four crucial capabilities for instruction tuning including complex reasoning, conversion, detail description, and referring question answering (Zhao et al., 2023). For each capability tuning, 20,000 samples are used for training while 5,000 samples are allocated for evaluation. **3) COAST-dataset**: Following (Chen et al., 2024), we integrate visual question-answering datasets including VQAv2 (Goyal et al., 2017), VizWiz (Gurari et al., 2018), ScienceQA (Lu et al., 2022), TextVQA (Singh et al., 2019), GQA (Hudson & Manning, 2019), OCR-VQA (Mishra et al., 2019), image classification dataset ImageNet (Deng et al., 2009), and referring expression comprehension dataset including RefCOCO (Kazemzadeh et al., 2014), RefCOCO+ (Mao et al., 2016) and RefCOCOg (Mao et al., 2016). Refer to (Chen et al., 2024) for the specific training and evaluation splits.

**Evaluation Metrics.** We customize the standard continual learning metrics (Wang et al., 2024; Chaudhry et al., 2018) for our continual instruction tuning scenario. We have set up two metrics for evaluation: 1) **average accuracy** represents the overall assessment of all the task performance. It is typically defined as the mean of the accuracy values obtained throughout all the tasks; 2) **average forgetting** aims to quantify the extent to which a model forgets previously learned tasks as it learns new ones. It is defined as the mean reduction between the maximum accuracy throughout the past learning process and the final accuracy. We follow (Liu et al., 2023b; Yin et al., 2024; Tong et al., 2024) to employ GPT-assisted assessment (we use GPT-4o (OpenAI, 2024) for grading) to evaluate the quality, relevance, and usefulness of model's predictions. Refer to Appendix A.1 for detailed explanations of the metrics and the grader prompt for GPT-4o.

**Compared Methods.** We consider the following methods for comparisons with Continual LLaVA: 1) *Sequential training* refers to the process of incrementally training a model on new tasks, where the model's parameters are initialized using weights pre-trained on previous tasks; 2) *Rehearsal training* involves the practice of replaying previously encountered data, often stored in a buffer, and integrating it with new tasks during the training process. Following (He et al., 2021; Huang et al., 2021), the buffer size is defined as 1% of the entire training task size; 3) *Popular continual learning methods* including regularization-based approaches (*i.e.*, EWC (Kirkpatrick et al., 2017) and LWF (Li & Hoiem, 2017)) and prompt-based methods (*i.e.*, L2P (Wang et al., 2022b), Dual (Wang et al., 2022a) and CODA (Smith et al., 2023)); 4) *Joint training* involves supplying the model with the full stream dataset simultaneously and training on all tasks collectively. This is typically regarded as the upper-bound performance of continual learning.

**Implementation Details.** We randomly sample three task orders from all the possible permutations of task compositions and report the mean results of average accuracy and average forgetting from the selected task orders. The specific task orders are available in Table 4 and Appendix A.1. The visual projector is implemented as two linear projection layers with a `GELU` activation function in between. The low-rank pool size $N$, the selected number $M$, and the rank number $R$ are respectively specified as 32, 4, and 8. We set the batch size to 32 and the learning rate $\eta$ to $4 \times 10^{-5}$ with a cosine decay schedule. The training process lasts for 2 epochs and the warm-up ratio is configured as 0.03. Following (Hu et al., 2021), the low-rank components $\boldsymbol{A}_n$ and $\boldsymbol{B}_n$ in Eq 1 are initialized with the zero and normal distribution, respectively.

Table 2: **Evaluation results (%) of continual instruction tuning on COAST-capability.** "Conv.", "Desc.", "Reason" and "Ref." represent conversation, detail description, complex reasoning, and referring qa, respectively. "Reh.", "Seq." and "Joint" denote rehearsal, sequential, and joint training.

| Methods | #Params | Avg.↑ | Fgt.↓ | Conv. | Desc. | Reason | Ref. |
|---------|---------|-------|-------|-------|-------|--------|------|
| Joint | 6.76B | 57.95 | — | 62.48 | 43.45 | 74.02 | 51.84 |
| CODA | 0.75M | 54.21 | 4.99 | 58.91 | 40.12 | 70.71 | 47.08 |
| Dual | 0.75M | 53.62 | 5.01 | 58.09 | 39.85 | 70.03 | 46.52 |
| L2P | 0.75M | 53.31 | 5.04 | 57.90 | 39.33 | 69.70 | 46.32 |
| LWF | 6.76B | 44.15 | 9.77 | 46.11 | 24.16 | 61.43 | 44.90 |
| EWC | 6.76B | 43.69 | 9.72 | 46.23 | 24.20 | 60.11 | 44.20 |
| Reh. | 6.76B | 43.34 | 9.79 | 45.11 | 23.93 | 60.54 | 43.76 |
| Seq. | 6.76B | 41.51 | 10.56 | 44.29 | 23.25 | 58.39 | 40.13 |
| Ours | **0.75M** | **55.79** | **4.18** | **60.42** | **41.25** | **72.25** | **49.23** |

Table 3: **Evaluation results (%) of continual instruction tuning on COAST-dataset.** "Reh.", "Seq." and "Joint" denote rehearsal, sequential, and joint training.

| Methods | Avg.↑ | Fgt.↓ | SciQA | Text | ImgNet | GQA | Viz | REC | VQA | OCR |
|---------|-------|-------|-------|------|--------|-----|-----|-----|-----|-----|
| Joint | 57.03 | — | 61.74 | 52.14 | 60.93 | 65.56 | 47.46 | 21.86 | 67.54 | 79.04 |
| CODA | 50.27 | 9.70 | 54.80 | 44.55 | 53.64 | 58.43 | 39.07 | 14.97 | 62.63 | 74.08 |
| Dual | 49.40 | 12.03 | 53.82 | 41.88 | 52.21 | 59.24 | 39.13 | 14.05 | 62.80 | 72.14 |
| L2P | 49.01 | 12.12 | 53.13 | 41.64 | 51.69 | 58.96 | 38.90 | 13.78 | 62.22 | 71.78 |
| LWF | 26.41 | 36.94 | 52.40 | 30.02 | 23.99 | 27.30 | 14.65 | 3.43 | 35.13 | 24.32 |
| EWC | 27.24 | 32.52 | 52.93 | 31.84 | 25.13 | 28.61 | 15.25 | 5.03 | 35.21 | 23.91 |
| Reh. | 26.49 | 33.17 | 52.02 | 31.29 | 24.44 | 28.03 | 14.80 | 4.14 | 34.14 | 23.03 |
| Seq. | 25.35 | 35.82 | 51.57 | 30.19 | 23.27 | 26.08 | 14.19 | 1.32 | 33.49 | 22.67 |
| Ours | **53.33** | **6.86** | **58.67** | **49.99** | **57.66** | **62.53** | **42.32** | **16.25** | **64.33** | **74.91** |

## 4.2 PERFORMANCE ANALYSIS

The experimental results for COAST-domain, COAST-capability and COAST-dataset are demonstrated in Table 1, Table 2 and Table 3, respectively. The comparisons highlight that Continual LLaVA consistently outperforms sequential training, rehearsal training, and leading continual learning methods in both average accuracy and average forgetting. For example, on COAST-domain, Continual LLaVA achieves an average accuracy of 37.08%, exceeding sequential training by a margin of 13.06%. Additionally, Continual LLaVA demonstrates a notably lower average forgetting than other approaches, further validating its ability to mitigate forgetting across different domains. Taking the sequential training and rehearsal training as examples, our approach reduced the forgetting rate by 13.25% (2.58% *v.s.* 15.83%) and 13.03% (2.58% *v.s.* 15.61%), respectively. Notably, our improvements come with the benefit of fewer tunable parameters. Our parameter-efficient tuning leverages only 0.75M tunable parameters, in stark contrast to the 7.67B parameters demanded by the sequential tuning. In summary, Continual LLaVA offers superior performance, less forgetting, and reduced computational overhead.

Through the comparisons under the domain, capability, and dataset incremental settings of COAST, we observe that the forgetting phenomenon of continual instruction learning is more pronounced on COAST-dataset. Specifically, the average forgetting of sequential training on COAST-dataset reaches 35.82%, respectively representing an absolute increase of 19.99% and 25.26% compared to the performance on COAST-domain and COAST-capability. The reason may lie in the steam datasets' highly diverse distributions and the ambiguity of task boundaries, which complicates LVLMs' ability to choose between retaining or revising previously acquired knowledge.

## 4.3 ABLATION STUDIES

**Ablations on the stream task order.** In Section 4.2, we present the average performance across three different task orders of COAST. Here, we aim to explore the impact of different task orders on continual instruction tuning. The results across different task orders are presented in Table 4 and

Table 4: **Ablation studies (%) on the task order**. We adopt the following abbreviation scheme to streamline the representation of task order notation. (a) On COAST-domain, `cdim` represents the order of **c**hart → **d**ocument → **i**con → **m**edical; (b) On COAST-capability, `crfd` denotes the order of **c**onv → **r**eason → re**f**qa → **d**esc; (c) On COAST-dataset, `stigzrvo` denotes the order of **S**ciQA → **T**ext → **I**mgNet → **G**QA → Vi**z** → **R**EC → **V**QA → **O**CR. Refer to Appendix A.1 for the explicit order referring to each abbreviation.

| Order | Avg.↑ | Fgt.↓ | Order | Avg.↑ | Fgt.↓ | Order | Avg.↑ | Fgt.↓ |
|---|---|---|---|---|---|---|---|---|
| cdim | 37.43 | 2.81 | crfd | 61.16 | 1.85 | stigzrvo | 52.67 | 8.22 |
| imcd | 36.65 | 2.62 | dcrf | 51.47 | 8.96 | vzgitosr | 53.62 | 4.95 |
| dmci | 37.17 | 2.30 | fdrc | 54.75 | 1.73 | itgzvors | 53.70 | 7.40 |

| (a) COAST-domain | (b) COAST-capability | (c) COAST-dataset |
|---|---|---|

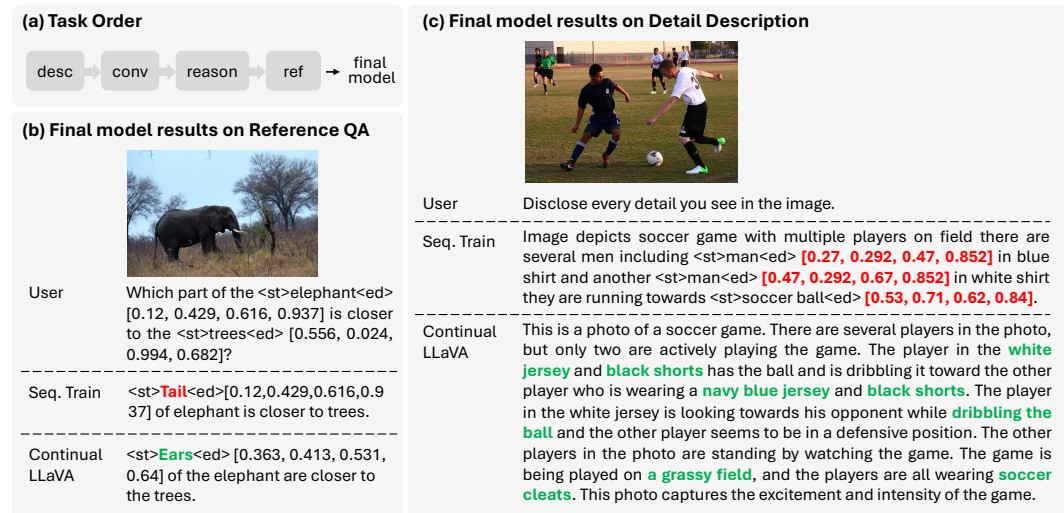

Figure 3: **Visualizations on reference QA and detail description tasks** under the training chain of `dcrf`, *i.e.*, desc → conv → reason → referring qa. The incorrect or undesired responses are marked in **red**, while the remarkable contents are highlighted in **green**.

the principal findings are as follows. **1)** In the context of COAST-domain, the task order does not significantly influence the final performance. This is likely attributable to the fact that each domain typically presents distinct patterns, resulting in minimal interference between tasks; **2)** For COAST-capability, the `dcrf` order yields a notably lower average accuracy of 51.47%, accompanied by a substantially high average forgetting of 8.96%. We conjecture that this phenomenon may stem from the fact that referring QA is designated as the final task to be learned in the `dcrf` order. This task focuses on a more specific localization capability and requires distinctive outputs with coordinates, potentially contributing to the forgetting of prior tasks. To further demonstrate this, we provide a visualization case in Figure 3. It shows that under the `dcrf` order, the final model of sequential training fails to retain the capability for `detail description` and invariably outputs unnecessary coordinate information. In contrast, our Continual LLaVA successfully differentiates between these two tasks and delivers accurate responses that align with the specified instructions.

**Ablations on dual increment embeddings.** We conduct ablation studies on the intrinsic and contextual increment embeddings to validate their contributions. The results in Table 5 (a) show that both intrinsic increment $\Delta\theta$ and contextual increment $\Delta\delta$ are crucial to the overall performance, *e.g.*, $\Delta\theta$ brings about 3.71% improvement in average accuracy and 0.25% decrease in average forgetting.

**Ablations on proxy-increment embedding alignment loss.** In Eq. 6, we align the selected proxy embeddings to the corresponding surrogate embeddings. We ablate on this alignment loss to see the difference and the comparison results are listed in Table 5 (b). We notice a significant 6.80% absolute decrease in average accuracy without applying the alignment loss, which demonstrates the necessity of aligning the proxy embeddings and surrogate embeddings.

Table 5: **Ablation studies (%) on (a) dual increment embeddings** including intrinsic increments $\Delta\theta$ and contextual increments $\Delta\delta$; **(b) the proxy-increment embedding alignment loss** $\mathcal{L}_{\text{align}}$; **(c) adaption positions** including the weight matrix of the `query`, `key`, `value` linear layers. "all-combination" denotes re-parameterizing all the `query`, `key`, `value`, and `output` linear layers; **(d) similarity computation mechanisms.** "vis-based sim" denotes mining intrinsic increments based on the similarity between visual embeddings and candidate embeddings.

| Exp. | Mode | Avg.↑ | Fgt.↓ | Chart | Doc. | Icon | Med. |
|---|---|---|---|---|---|---|---|
| | vanilla | **37.08** | **2.58** | 15.30 | **17.82** | **60.71** | **54.50** |
| (a) | _w/o_ $\Delta\theta$ | $33.37_{-3.71}$ | $2.83_{+0.25}$ | 11.92 | 14.11 | 56.87 | 50.59 |
| | _w/o_ $\Delta\delta$ | $36.43_{-0.65}$ | $2.89_{+0.31}$ | 15.04 | 17.10 | 59.94 | 53.62 |
| (b) | _w/o_ $\mathcal{L}_{\text{align}}$ | $30.28_{-6.80}$ | $2.91_{+0.33}$ | 13.13 | 15.97 | 51.56 | 40.50 |
| (c) | query-adaption | $36.41_{-0.67}$ | $2.65_{+0.07}$ | 14.96 | 17.04 | 59.90 | 53.74 |
| | key-adaption | $36.42_{-0.66}$ | $2.65_{+0.07}$ | 14.98 | 16.99 | 59.93 | 53.76 |
| | value-adaption | $36.43_{-0.65}$ | $2.65_{+0.07}$ | 15.02 | 17.02 | 59.91 | 53.78 |
| | all-adaption | $36.99_{-0.09}$ | $2.62_{+0.04}$ | **15.31** | 17.65 | 60.62 | 54.38 |
| (d) | vis-based sim | $35.67_{-1.41}$ | $2.77_{+0.19}$ | 13.75 | 16.15 | 58.82 | 53.94 |

Table 6: **Hyper-parameter ablations** of (a) low-rank pool size $N$ and (b) selected number $M$.

| $N$ | 8 | 16 | 32 | 64 | | $M$ | 1 | 4 | 8 | 16 |
|---|---|---|---|---|---|---|---|---|---|---|
| **Avg.↑** | 34.04 | 35.13 | **37.08** | 37.06 | | **Avg.↑** | 35.12 | **37.08** | 36.97 | 36.82 |
| **Fgt.↓** | 2.89 | 2.62 | **2.58** | 2.59 | | **Fgt.↓** | 2.92 | **2.58** | 2.62 | 2.65 |

(a) The low-rank pool size $N$.  (b) The selected number $M$.

**Ablations on adaption positions.** In Sec. 3.3, we adapt the constructed dual increment embeddings into the `output` linear layer. We conduct ablation experiments on the adaption positions, including the linear layer of `query`, `key`, `value`, `output`, and their combination. Refer to Figure 4 in Appendix for schematic illustrations. The comparison results are listed in Table 5 (c). We have the following findings: **1)** The performance of query-adaptation, key-adaptation, and value-adaptation are comparable, but all fall short in comparison to output-adaptation in vanilla Continual LLaVA; **2)** Re-parameterizing all four linear layers is unnecessary since the "all-adaption" results are inferior to that of output-adaption. Therefore, we opted for "output-adaption" for re-parameterization.

**Ablations of similarity computation mechanisms.** In Sec. 3.2, the intrinsic increment embeddings are mined based on the cosine similarity between the textual instruction and proxy embeddings, _i.e._, _text-based_ similarity. Here we ablate on the selection manner according to the _vision-based_ similarity, _i.e._, the cosine similarity between visual embeddings and candidate proxy embeddings. Specifically, the visual embeddings are extracted by a pre-trained CLIP visual encoder ViT-L/14 (Radford et al., 2021). The results in Table 5 (d) demonstrate that vision-based selection leads to inferior performance, which may be due to the fact that textual instructions more easily differentiate between tasks and provide explicit task objectives.

**Ablations of hyper-parameters.** We conduct hyper-parameter ablation studies including low-rank pool size $N$ and selected number $M$ on COAST-domain. According to the results in Table 6, we set $N = 32$ and $M = 4$ for the optimum performance.

## 5 CONCLUSIONS

This paper targets continual instruction tuning, which refers to the process of incrementally adapting LVLM to new tasks by fine-tuning it with task-specific instructions. To establish an assessment standard, we propose COAST as the benchmark for continual instruction tuning on LVLMs from the domain-incremental, capability-incremental, and dataset-incremental perspectives. In addition, we propose a parameter-efficient tuning method Continual LLaVA, which devises the intrinsic increment embeddings to capture task-specific properties and contextual increment embeddings to explore inter-task relational dependencies. Experimental results manifest that Continual LLaVA significantly improves the overall performance and reduces catastrophic forgetting during the continual instruction tuning process.

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

# A APPENDIX

This appendix contains the additional details including the following aspects:

- More Details of Experimental Settings (Sec. A.1)
  - Evaluation prompt for COAST
  - Task order reference
  - Illustrations of evaluation metrics
  - Illustrations of adaption positions
  - Illustrations of grade prompt for GPT
  - Algorithm for inference
- More Experimental Results (Sec. A.2)
  - Specific results of Continual LLaVA on each task order
  - Plug-and-play analysis
  - Comparisons of on-the-fly results and final model results
  - Ablations on the low-rank decomposition
- More Related Work Discussion (Sec. A.3)
  - Continual learning for LLMs
  - LVLM benchmarks
- More Visualization Results (Sec. A.4)
  - Visualizations of low-rank pool selection
  - Visualizations of training losses
  - Qualitative comparisons between sequential training and Continual LLaVA
- **Source Codes and Reproducibility** (Sec. A.5)

## A.1 MORE DETAILS OF EXPERIMENTAL SETTINGS

**Evaluation Prompt for COAST.** Following (Tong et al., 2024), the prompts used for COAST benchmark evaluation are released in Table 7. For datasets that are not explicitly designated, no additional evaluation prompts are applied.

**Task Order Reference:** In Table 4, we conduct ablations on three different task orders. Here we provide the specific task order sequence of the task abbreviation for more convenient reference.

The task order reference on COAST-domain is as follows:

- **cdim**: **c**hart → **d**ocument → **i**con → **m**edical
- **imcd**: **i**con → **m**edical → **c**hart → **d**ocument
- **dmci**: **d**ocument → **m**edical → **c**hart → **i**con

The task order reference on COAST-capability is as follows:

- **crfd**: **c**onversation → complex **r**eason → re**f**erring qa → detail **d**escription
- **dcrf**: detail **d**escription → **c**onversation → complex **r**eason → re**f**erring qa
- **fdrc**: re**f**erring qa → detail **d**escription → complex **r**eason → **c**onversation

The task order reference on COAST-dataset is as follows:

- **stigzrvo**: **S**ciQA → **T**ext → **I**mgNet → **G**QA → Vi**z** → **R**EC → **V**QA → **O**CR.
- **vzgitosr**: **V**QA → Vi**z** → **G**QA → **I**mgNet → **T**ext → **O**CR → **S**ciQA → **R**EC
- **itgzvors**: **I**mgNet → **T**ext → **G**QA → Vi**z** → **V**QA → **O**CR → **R**EC → **S**ciQA

Table 7: **Prompts used in the evaluation** for the related datasets.

| Dataset | Prompt | Example |
|---|---|---|
| ChartQA | \nAnswer the question using a single number or phrase. | <image>\nWhat was the sales volume of computers and telecoms in the second quarter of 2020?\nAnswer the question using a single number or phrase. |
| DocVQA | \nGive the short answer directly. | <image>\nWhat is the time of the Seminar?\nGive the short answer directly. |
| IconQA | \nAnswer with the option letter from the given choices directly. | <image>\nHow many shapes are blue?\nAnswer with the option letter from the given choices directly. |
| MedicalQA | \nAnswer the question using a single word or phrase. | <image>\nIs tuberculous peritonitis present?\nAnswer the question using a single word or phrase. |
| ScienceQA | \nAnswer with the option's letter from the given choices directly. | <image>\nWhen World War I first started, what did many people believe?\nA. It would be one of the longest wars in history.\nB. The war would be the first of two world wars.\nC. The war would lead to the death of millions of Germans.\nD. The war would be over quickly.\nAnswer with the option's letter from the given choices directly. |
| Text-VQA | \nAnswer the question using a single word or phrase. | <image>\nHow man price tags are on the bottom shelf?\nReference OCR tokens: 2.39, 2.45, 2.39, 2.39, 39\nAnswer the question using a single word or phrase. |
| ImageNet | \nAnswer the question using a single word or phrase. | <image>\nWhat is the object in the image? \nAnswer the question using a single word or phrase. |
| GQA | \nAnswer the question using a single word or phrase. | <image>\nIs the sky dark?\nAnswer the question using a single word or phrase. |
| VizWiz | \nAnswer the question using a single word or phrase. | <image>\nWhat's the name of this product?\nAnswer the question using a single word or phrase. |
| VQAv2 | \nAnswer the question using a single word or phrase. | <image>\nWhat is this photo taken looking through?\nAnswer the question using a single word or phrase. |

**Evaluation Metrics of Continual Instruction Tuning.** We devise the metrics of *average accuracy* and *average forgetting* used to evaluate the continual instruction tuning performance. The former represents the overall performance of the final model on all the learned tasks while the latter measures how much the model's performance on older tasks has degraded as it learns new ones.

Let $\alpha_{k,j} \in [0, 1]$ denote the GPT-evaluated accuracy on $j$-th task after incrementally training on the $k$ sequential tasks ($j \leq k$). The metric of average accuracy is defined as the mean values of GPT-evaluated accuracy of the final model across all the learned tasks.

$$\text{AA}_k = \frac{1}{k} \sum_{j=1}^{k} a_{k,j}. \tag{7}$$

Since average accuracy does not convey any insight into the forgetting dynamics during the continual instruction tuning process, average forgetting has been introduced to fill this gap. For a particular task, the forgetting measure is defined as the difference between the maximum accuracy throughout the past learning process and the current one. In particular, the forgetting for the $j$-th task after incrementally training up to $k$ tasks is as follows.

$$f_j^k = \max_{l \in \{1, \cdots, k-1\}} a_{l,j} - a_{k,j}, \quad \forall j < k. \tag{8}$$

The average forgetting of $k$-th task is computed as follows.

$$\text{AF}_k = \frac{1}{k-1} \sum_{j=1}^{k-1} f_j^k. \tag{9}$$

We report the average accuracy and average forgetting after learning across all the $T$ tasks, $i.e.$, $\text{AA}_T$ and $\text{AF}_T$.

**Illustrations on Adaption Positions.** Recall that after obtaining the intrinsic and contextual embeddings, we adapt them into the linear projection layers of LLM. There exist four choices including the `query`, `key`, `value`, and `output` projections. The schematic illustration of the adaption positions is demonstrated in Figure 4. According to the comparison experiments in Table 5(c), we opt to adapt the constructed increment embeddings into the `output` linear layer.

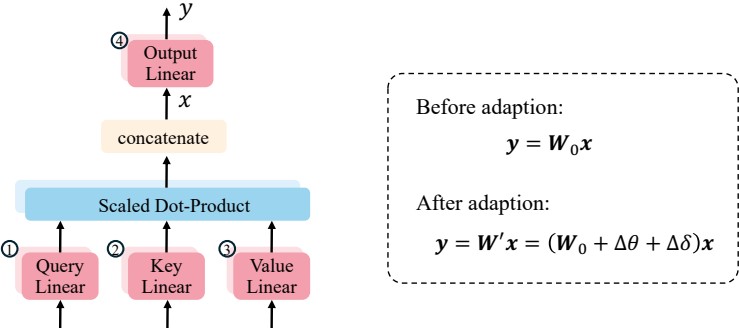

Figure 4: **Illustrations of adaption positions** including the `query`, `key`, `value`, and `output` linear projections. $\Delta\theta$ and $\Delta\delta$ denote intrinsic and contextual increment embeddings, respectively.

**Grade Prompt.** We follow (Liu et al., 2023b; Yin et al., 2024; Tong et al., 2024) to employ GPT-assisted assessment to evaluate the quality of model predictions. We choose GPT-4o and the grader prompts are as follows.

> **System prompt for LLM Grader**
>
> ```
> You are an intelligent chatbot designed for evaluating the
> correctness of generative outputs for question-answer pairs.
> Your task is to compare the predicted answer with the correct
> answer and determine if they match meaningfully. Here's how
> you can accomplish the task:
> ------
> ##INSTRUCTIONS:
> - Focus on the meaningful match between the predicted answer
> and the correct answer.
> - Consider synonyms or paraphrases as valid matches.
> - Evaluate the correctness of the prediction compared to
> the answer.
> ```

**Algorithm for Inference.** We provide the algorithm for inference in Algorithm 2. Notably, the inference process does not depend on experience replay or task-specific identification.

## A.2   MORE EXPERIMENTAL RESULTS

**Specific Results for Each Task Order.** In Table 4, we report the average accuracy and average forgetting under different task orders on the COAST benchmark. Here we augment Table 4 by providing the specific performance on each task. The performance of Continual LLaVA on COAST-domain, COAST-capability, and COAST-dataset under different task orders are listed in Table 8.

**Plug-and-play Analysis.** Our proposed dual increment embedding mining can serve as the plug-and-play strategy that can be easily applied to other LVLMs. Besides the LLaVA (Liu et al., 2023a)

---

**Algorithm 2** The inference pipeline of Continual LLaVA

---

**Input:** Image $v_t^i$, textual instructions $s_t^i$.
**Output:** Responses $r_t^i$.

1: **function** INFER($v_t^i, s_t^i$)
2:     Extract surrogate feature $q_t^i = \text{Sentence-BERT}(s_t^i)$
3:     Compute cosine similarities between $q_t^i$ and proxy feature $k_n$ as $\cos(q_t^i, k_n)$
4:     Obtain index set $\mathcal{I} = \{i_1, i_2, \cdots, i_M\}$ with top-$M$ highest similarities via Eq. 2
5:     Compute intrinsic increment embedding $\Delta\theta_t^i \leftarrow \frac{\sum_{m=1}^{M} \cos(q_t^i, k_{i_m}) \cdot P_{i_m}}{\sum_{m=1}^{M} \cos(q_t^i, k_{i_m})}$
6:     Compute contextual increment embedding $\Delta\delta_t^i \leftarrow \sum_{l=1}^{T} w_l \cdot \overline{\mathcal{Z}}_t$
7:     Re-parameterize LLM via Eq. 5 and generate responses $r_t^i$
8:     **return** $r_t^i$
9: **end function**

---

Table 8: **Performance (%) of Continual LLaVA on COAST benchmark under different task orders.** The information related to the abbreviation of task order can be accessed in Sec.A.1.

| Order | Avg.↑ | Fgt.↓ | Chart | Doc. | Icon | Med. |
|---|---|---|---|---|---|---|
| cdim | 37.43 | 2.81 | 14.05 | 17.78 | 61.63 | 56.27 |
| imcd | 36.65 | 2.62 | 16.27 | 18.76 | 56.98 | 54.57 |
| cdim | 37.17 | 2.30 | 15.58 | 16.91 | 63.53 | 52.66 |

(a) COAST-domain

| Order | Avg.↑ | Fgt.↓ | Conv | Desc | Reason | Ref |
|---|---|---|---|---|---|---|
| crfd | 61.16 | 1.85 | 66.20 | 51.86 | 82.14 | 44.42 |
| dcrf | 51.47 | 8.96 | 56.82 | 31.18 | 66.84 | 51.02 |
| fdrc | 54.75 | 1.73 | 58.24 | 40.72 | 67.78 | 52.24 |

(b) COAST-capability

| Methods | Avg.↑ | Fgt.↓ | SciQA | Text | ImgNet | GQA | Viz | REC | VQA | OCR |
|---|---|---|---|---|---|---|---|---|---|---|
| stigzrvo | 52.67 | 8.22 | 54.78 | 48.16 | 81.30 | 60.56 | 36.48 | 2.086 | 63.26 | 74.74 |
| vzgitosr | 53.62 | 4.95 | 61.43 | 50.10 | 44.86 | 63.54 | 46.86 | 24.12 | 62.90 | 75.18 |
| itgzvors | 53.70 | 7.40 | 59.79 | 51.70 | 46.82 | 63.50 | 43.62 | 22.54 | 66.84 | 74.82 |

(c) COAST-dataset

architecture employed in the main paper, we also experiment based on MiniGPT-4 (Zhu et al., 2023a). The results on COAST-domain benchmark are demonstrated in Table 9. The comparison results indicate that our proposed intrinsic and contextual increments are also effective based on the MiniGPT-4 architecture, demonstrating the generalizability of the proposed dual increment embeddings.

**Visualization of Forgetting.** We seek to clearly demonstrate how *forgetting* arises during the continual instruction tuning process, thereby further emphasizing the necessity and significance of exploring continual learning in the context of instruction tuning. To this end, we visualize both the *on-the-fly accuracy* and the *final model accuracy*. The former represents the snapshot performance of the model trained on a new task and then evaluated immediately on that task before moving to the next. The latter denotes the performance of continually training the model on the task stream and is evaluated after finishing the training of the last task.

We compare the naive sequential training and the proposed Continual LLaVA on both the metrics of on-the-fly accuracy and the final model accuracy. We report the results on COAST-capability under three different training orders. The comparisons are depicted in Figure 5 and we can draw the following conclusions: **1)** The phenomenon of forgetting frequently occurs during continual instruction tuning. For example in Figure 5b, there exists a 32.80% performance gap (50.02% *v.s.* 17.22%) between the on-the-fly accuracy and the final model accuracy on the conversation task. This stresses the importance of advancing research on continual learning for instruction tuning; **2)** Our

Table 9: **Plug-and-play analysis (%)** of the proposed dual increment embeddings on COAST-domain. We adapt the constructed intrinsic and contextual increment embeddings into LLaVA (Liu et al., 2023a) and MiniGPT-4 (Zhu et al., 2023b), respectively.

| Method | Avg.↑ | Fgt.↓ | Chart | Doc. | Icon | Med. |
|---|---|---|---|---|---|---|
| LLaVA Sequential | 24.02 | 15.83 | 11.77 | 11.29 | 33.73 | 39.27 |
| + dual increments | **37.08**$_{+13.06}$ | **2.58**$_{-13.25}$ | **15.30** | **17.82** | **60.71** | **54.50** |
| MiniGPT-4 Sequential | 28.65 | 9.30 | 11.60 | 11.77 | 44.91 | 46.32 |
| + dual increments | **31.02**$_{+2.37}$ | **3.43**$_{-5.87}$ | **12.45** | **14.04** | **49.66** | **47.93** |

Table 10: **Ablations (%) on the low rank decomposition** for increment embedding generation.

| Method | Avg.↑ | Fgt.↓ | Chart | Doc. | Icon | Med. |
|---|---|---|---|---|---|---|
| *w/ low-rank* | **37.08** | **2.58** | **15.30** | **17.82** | **60.71** | **54.50** |
| *w/o low-rank* | 36.21$_{-0.87}$ | 2.80$_{+0.22}$ | 14.11 | 16.73 | 60.02 | 53.99 |

proposed Continual LLaVA can substantially mitigate the forgetting phenomenon. For example in the conversation task of Figure 5b, Continual LLaVA reduces the performance gap between the one-the-fly accuracy and the final model accuracy to 3.14%; **3)** Notably, the final accuracy of Continual LLaVA in certain cases exceeds that of on-the-fly accuracy, *e.g.*, the complex reasoning task in Figure 5a. This highlights that our approach can better capitalize on the interdependencies among tasks to enhance the performance of previously acquired tasks.

**Ablations on the Low-rank Decomposition** In Eq 5, the increment embeddings $P_n$ are generated following the low-rank spirit. Instead, we conduct ablation experiments by directly initializing $P_n \in \mathbb{R}^{d \times d}$ without using the low-rank decomposition. The comparison experiments are summarized in Table 10, which demonstrates the advantages of utilizing low-rank decomposition (37.08% *v.s.* 36.21% in average accuracy) in parameter efficient tuning.

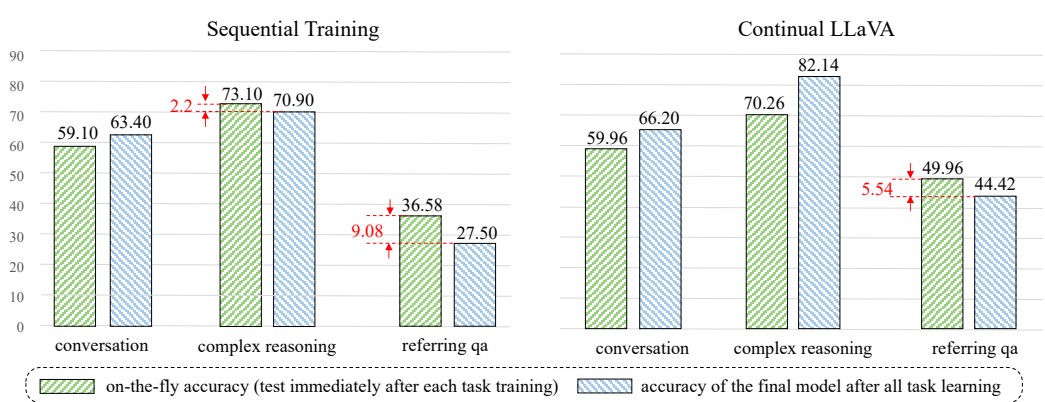

(a) Comparisons of on-the-fly accuracy and final model accuracy for sequential training (left) and our Continual LLaVA (right) under the task of conv → reason → ref → desc.

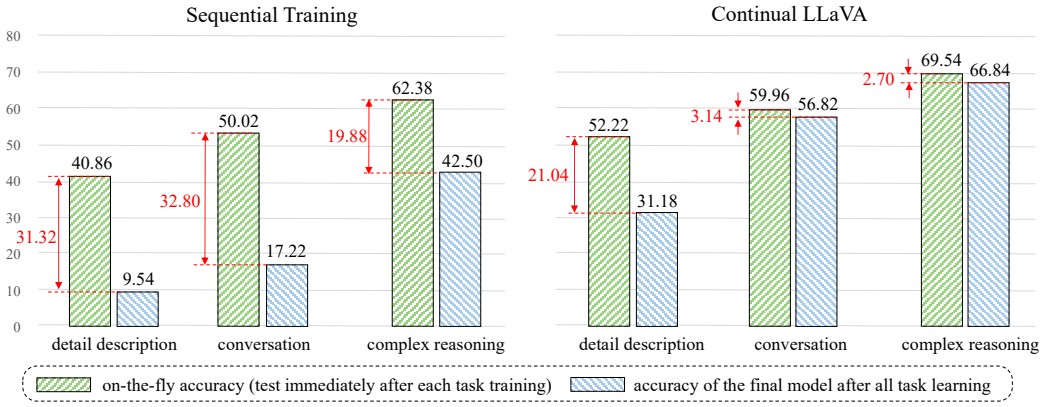

(b) Comparisons of on-the-fly accuracy and final model accuracy for sequential training (left) and our Continual LLaVA (right) under the task of desc → conv → reason → ref.

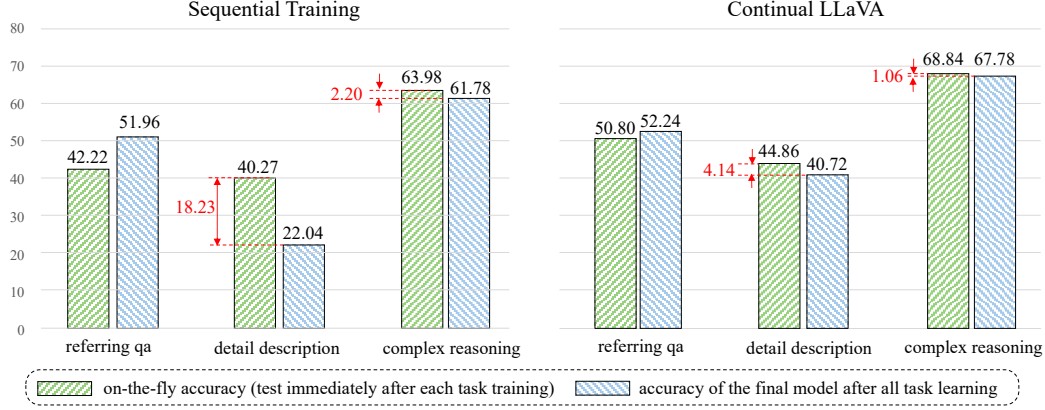

(c) Comparisons of on-the-fly accuracy and final model accuracy for sequential training (left) and our Continual LLaVA (right) under the task order of ref → desc → reason → conv.

Figure 5: **Visualization of forgetting (%)** on each task for sequential training (left) and our Continual LLaVA (right) under different task orders.

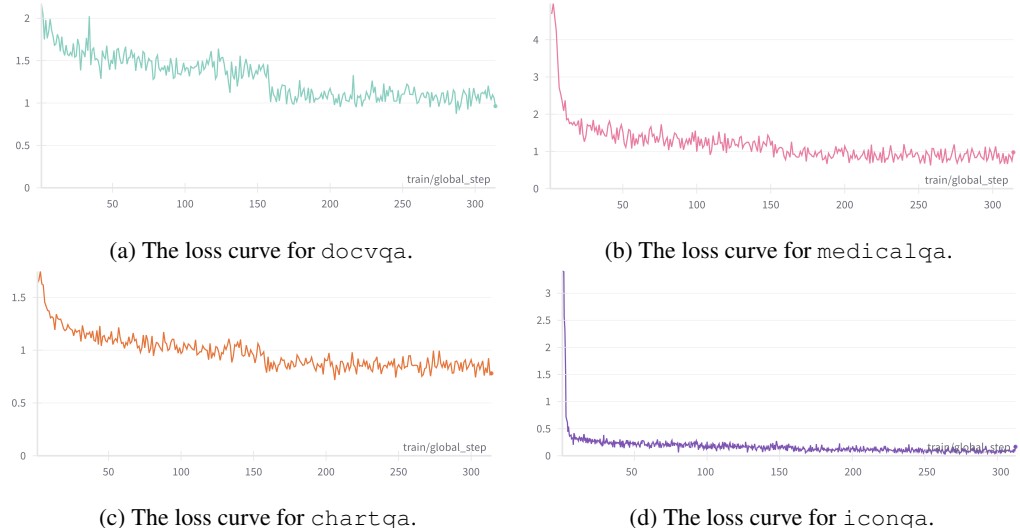

(a) The loss curve for `docvqa`.

(b) The loss curve for `medicalqa`.

(c) The loss curve for `chartqa`.

(d) The loss curve for `iconqa`.

Figure 6: **Visualizations of the training loss curves** of Continual LLaVA on COAST-domain benchmark. The training order is set to `document → medical → chart → icon`.

### A.3 MORE RELATED WORK DISCUSSION

**Continual Learning for LLMs.** Due to the massive parameter scale and complexity, continual learning for LLMs encounters multi-faceted challenges (Shi et al., 2024; Wu et al., 2024b). Based on the training process of LLMs, continual learning for LLMs (Bohao et al., 2024; Jin et al., 2022; Razdaibiedina et al., 2023) can be classified into three fields including continual pre-training, continual instruction tuning, and continual preference alignment.

Continual pre-training (Jin et al., 2022; Jang et al., 2022a; Ke et al., 2023) aims to incorporate updated world knowledge into LLMs by training them on extensive and diverse datasets. A prevalent application of continual pretraining involves dynamically gathering data from multiple sources including news feeds (Sun et al., 2020) and scholarly articles (Cossu et al., 2024), enabling LLMs to stay aligned with up-to-date information (Jang et al., 2022b;a). Other methods tailor LLMs to specific fields via continual pre-training. (Xie et al., 2023) adapts LLMs into the financial understanding and EcomGPT-CT (Ma et al., 2023) investigates continual pre-training in the E-commerce domain. (Gogoulou et al., 2023) enhances LLMs' ability to understand regional dialects and contemporary slangs across diverse social and cultural groups.

Continual instruction tuning (Zhang et al., 2023c; Wang et al., 2023b;a; Zhao et al., 2024) continuously finetunes LLMs on a sequence of task-specific instructions and develops the competence to address emerging tasks. ProgPrompt (Razdaibiedina et al., 2023) keeps most parameters of LLMs frozen and only trains a fixed set of prompt tokens for each new task. To alleviate the reliance on inference task-ID, SLM (Bohao et al., 2024) proposes a task-related knowledge retrieval technique to enable adaptive adjustment for downstream tasks. LLaMA Pro (Wu et al., 2024a) expands the block within LLMs to facilitate the knowledge injection into LLMs and obtain the trade-off between general knowledge and domain-specific capabilities.

Continual preference alignment (Zhang et al., 2023a; Yao et al., 2023) adapts LLMs to evolving societal values and ethical guidelines. The typical methodology is reinforcement learning with human feedback (RLHF) (Kaufmann et al., 2023), which combines principles of reinforcement learning with feedback from human evaluators to improve the alignment with human preferences and values. The follow-up work CPPO (Zhang et al., 2024) enhances Proximal Policy Optimization (PPO) (Schulman et al., 2017) algorithm with instance-wise weights to balance policy exploration and knowledge retention. (Zhang et al., 2023a) extends the Direct Preference Optimization (DPO) algorithm (Rafailov et al., 2024) by employing Monte Carlo estimation (Harrison, 2010) to derive optimal policy sequences for stream tasks.

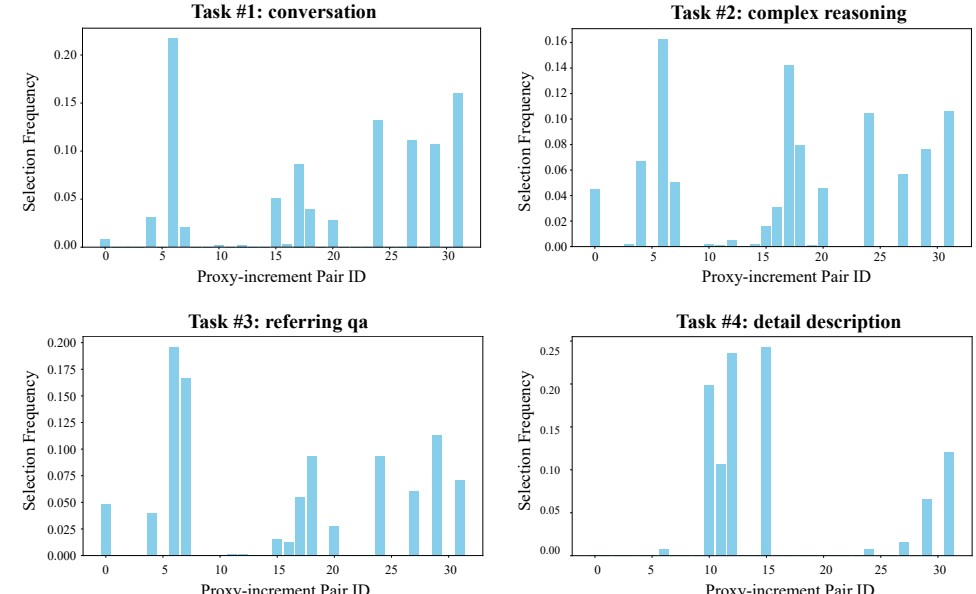

Figure 7: **Visualizations of the increment embedding selection frequency on COAST-capability.** The task order is set to `conv → reason → refqa → desc`.

**LVLM Benchmarks.** With the advent of comprehensive LVLMs (Liu et al., 2023a; OpenAI, 2024), a wide range of evaluation benchmarks (Liu et al., 2023b; Yu et al., 2023; Huang & Zhang, 2024) have been introduced to assess their performance across various dimensions. Based on the model competencies being examined, LVLM benchmarks can be classified into two categories including *general* capabilities for multi-modal understanding and *specific* capabilities for downstream applications. Typical general-purpose LVLM Benchmarks include MMBench (Liu et al., 2023b), MM-Vet (Yu et al., 2023), Seed-Bench (Li et al., 2023a), *etc.*, with the focus on multi-modal perception (*e.g.*, recognition or localization) and reasoning (commonsense or logic reasoning). The specific capabilities involve natural science (*e.g.*, ScienceQA (Lu et al., 2022), MathVista (Lu et al., 2023)), medical usage (*e.g.*, MMMU (Yue et al., 2024), M3D (Bai et al., 2024)), agent planning (*e.g.*, OpenEQA (Majumdar et al., 2024)), remote sensing (*e.g.*, RSGPT (Hu et al., 2023)), *etc.* Most of the current benchmarks focus on the single-task adaption of LVLMs and neglect the consistent adaption among different tasks. To facilitate continual instruction tuning, we propose COAST by chaining and re-purposing current benchmarks in a stream format.

## A.4   MORE VISUALIZATIONS

**Visualization of Training Loss.** We plot the training loss of Continual LLaVA on the COAST-domain benchmark in Figure 6. Specifically, we visualize the training loss under the training order of `document → medical → chart → icon`.

**Visualizations of Increment Embedding Selection.** We employ cosine similarity as the measurement between the input instruction and proxy-increment pairs within the low-rank pool, from which we select the top-$M$ increment embeddings. Figure 7 illustrates the selection frequency of each proxy-increment pair within the pool during the training phase of COAST-capability.

**More Visualization Results.** We provide more qualitative comparisons between the sequential training and Continual LLaVA. The visualization results in Table 11–20 demonstrate the performance of the proposed Continual LLaVA.

## A.5   SOURCE CODES AND REPRODUCIBILITY

**We include the source codes of Continual LLaVA in the supplementary material.** Refer to the contained "README.md" file for reproducibility.

Table 11: Comparison results between sequential training and our Continual LLaVA on the task of detail description.

| | |
|---|---|
| **Training order:** detail description → conversation → complex reason → referring qa | |
| **Evaluation results on the task of detail description:** | |

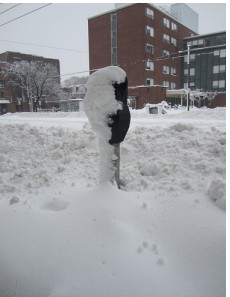

| | |
|---|---|
| User | Could you give a blow-by-blow description of the image? |
| Sequential Train | Image shows scene with <st>banana<ed> [0.396 0.588 0.566 0.782] that has been cut in half placed on <st>wooden bowl<ed> [0.004 0.588 0.998 0.998] which is on ground there are <st>green leaves<ed>[0.004 0.002 0.998 0.821] |
| Continual LLaVA | The image captures a moment of quiet contemplation in the heart of a lush tropical garden. A small brown lizard, perhaps a gecko, perches precariously on the edge of a large, round, dark grey object, likely a stone trough or basin. It's looking to the left, its body angled towards the viewer, its long tail trailing down towards the ground. The lizard's focus is on something outside the frame, a testament to its keen senses and alert nature. The trough is partially filled with water, reflecting the dappled sunlight filtering through the foliage. Two half-eaten banana peels rest on the object's surface, offering a tempting suggestion of recent activity in this quiet corner. The background is a symphony of vibrant green, a blurred tapestry of leaves and stems, hinting at the rich biodiversity of the surrounding jungle. The soft focus and subtle lighting create a sense of tranquility, making the lizard's presence all the more striking. |

[*Start a new conversation, and clear the history*]

| | |
|---|---|
| User | Can you portray the image in words? |
| Sequential Train | Image portrays snowy scene with <st>parking meter<ed> [0.396, 0.283, 0.566, 0.997] covered in snow there are <st>buildings<ed> [0.004, 0.003, 0.998, 0.667] in background and sky is <st>gray and overcast<ed> [0.004, 0.003, 0.998, 0.732]. |
| Continual LLaVA | The image portrays a snow-covered cityscape engulfed in the midst of a fierce winter storm. A thick layer of snow blankets the streets, rising nearly halfway up parking meters and leaving deep impressions where footprints dot the otherwise undisturbed expanse. The city's main road is largely covered in snow, with only a small section cleared, hinting at recent traffic or attempts to maintain passage. By the roadside, a partially obscured stop sign and a snow-laden bus stop point to the city's public transport system. The scene is dominated by a large, brick-red building, where scaffolding atop suggests ongoing construction or renovation. A bare tree, dusted with snow, stands next to the road, adding to the wintry charm. Above, power lines stretch across the street, connecting parts of the city through the snowstorm. Small details, like a snowball left behind and a white overhang near the building's entrance, add life to the scene. A grey, overcast sky looms above, casting a peaceful yet melancholic mood over the transformed urban landscape. |

Table 12: Comparison results between sequential training and our Continual LLaVA on the task of detail description.

---

**Training order:** detail description → conversation → complex reason → referring qa
**Evaluation results on the task of detail description:**

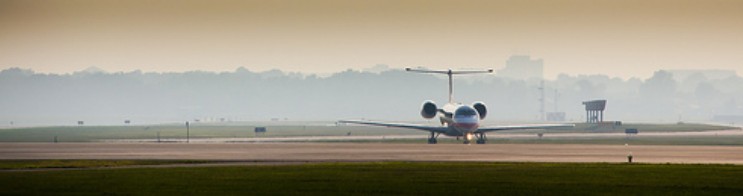

| User | Disclose every detail you see in the image. |
|------|---------------------------------------------|
| Sequential Train | I see <st>tree<ed> [0.003, 0.003, 0.21, 0.667] with <st>leaves<ed> [0.003, 0.003, 0.21, 0.667], <st>sidewalk<ed> [0.003, 0.667, 0.998, 0.997] and <st>car<ed> [0.003, 0.423, 0.215, 0.50]. |
| Continual LLaVA | The image features two parking meters mounted on a metal pole, surrounded by a small patch of greenery bordered by a short iron fence. Various plants and shrubs grow around the base, creating a contrast between urban infrastructure and nature. Behind the meters, parked cars are partially visible, indicating a city or suburban street setting. In the background, trees and buildings suggest a peaceful neighborhood scene, with a few distant people and cars adding subtle activity to the image. A large tree on the right side of the image blends with the surrounding greenery. The warm sunlight illuminates the scene, casting a golden glow across the area, while a soft lens flare appears near the right edge, adding a dreamy, nostalgic feel to the composition. The image captures a harmonious balance between urban elements and natural beauty. |

---

[*Start a new conversation, and clear the history*]

| User | Go into detail about the different elements of the image. |
|------|-----------------------------------------------------------|
| Sequential Train | Image features <st>plane<ed> [0.356, 0.452, 0.724, 0.652] on <st>runway<ed> [0.004, 0.612, 0.998, 0.997] with <st>tarmac<ed> [0.004, 0.612, 0.998, 0.997] beneath it there are <st>trees<ed> |
| Continual LLaVA | The image is a panoramic shot of an airplane on a runway, taken on a hazy day. The plane is a regional jet, painted in a white and red livery. It is positioned on the runway, facing the camera. The plane is positioned on the runway, facing the camera, with the front landing gear down. The plane's engines are visible, and the tail is slightly angled up. The runway is surrounded by a grassy field, and a line of trees in the background. The sky is a muted, pale color, suggesting the time of day is either early morning or late evening. The image is well-composed and well-lit, capturing the plane in a moment of quiet anticipation. It evokes a sense of stillness and peace, suggesting a moment before the plane takes off. |

---

Table 13: Comparison results between sequential training and our Continual LLaVA on the task of detail description.

---

**Training order:** detail description → conversation → complex reason → referring qa
**Evaluation results on the task of detail description:**

---

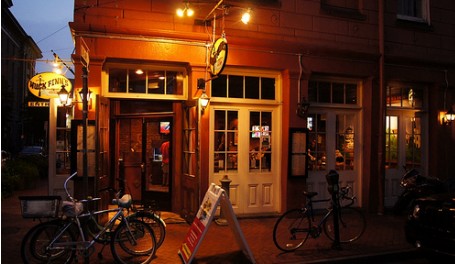

| | |
|---|---|
| User | Offer an exhaustive description of the image. |
| Sequential Train | Image depicts large ornate <st>building<ed> [0.002, 0.454, 0.932, 0.433] with <st>clock tower<ed> [0.23, 0.433, 0.762, 0.783] situated against backdrop of <st>blue sky<ed> [0.322, 0.235, 0.855, 0.892]. |
| Continual LLaVA | The image captures a majestic brick building, its architecture hinting at a historical significance. The building features a central tower, crowned with a spire and topped with a clock. This tower dominates the composition, rising against a clear, vibrant blue sky. The building is adorned with arched windows, showcasing a meticulous attention to detail. It is bathed in warm sunlight, highlighting the red tones of the brick and casting a gentle glow on the stonework. A small, white moon hangs in the sky, adding a touch of celestial beauty to the scene. The overall mood is serene and dignified, suggesting a place of history and perhaps even civic importance. The building's prominence against the vast expanse of sky creates a sense of grandeur and timeless elegance. |

---

[*Start a new conversation, and clear the history*]

| | |
|---|---|
| User | Kindly provide a meticulous commentary on the image. |
| Sequential Train | Image depicts scene during night with <st>red brick building<ed> [0.003 0.003 0.998 0.997] with <st>red brick wall<ed> [0.003 0.003 0.998 0.997] there are several <st>bicycles<ed> [0.003 0.643 0.998 0.997]. |
| Continual LLaVA | The image captures a lively, bustling scene outside a bar, likely in a city setting. The focus is on the storefront of a bar called "Mucky Finn's," located on a street corner. The bar's exterior is a warm, terracotta color, illuminated by the golden glow of streetlights and warm interior lighting. The building has a classic, old-world charm, with large windows that showcase the activity within. A pair of bikes, locked together, stand outside the bar, a testament to the establishment's popularity as a social hub. The bikes add a touch of urban cool to the scene, hinting at the energy and vibrancy of the surrounding neighborhood. In the background, another storefront is visible, further suggesting that this is a lively street with a vibrant nightlife. The warm lighting, the vibrant colors, and the bustling activity all contribute to a feeling of warmth and excitement, inviting viewers to imagine stepping into this lively setting. |

---

Table 14: Comparison results between sequential training and our Continual LLaVA on the task of ScienceQA.

**Training order:** SciQA → Text → ImgNet → GQA → Viz → REC → VQA → OCR
**Evaluation results on the task of ScienceQA:**

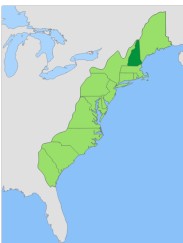

| | |
|---|---|
| User | What is the name of the colony shown?
A. Maryland
B. New Hampshire
C. Rhode Island
D. Vermont
Answer with the option's letter from the given choices directly. |
| Sequential Train | New Hampshire. |
| Continual LLaVA | B |
| Ground Truth | B |

[*Start a new conversation, and clear the history*]

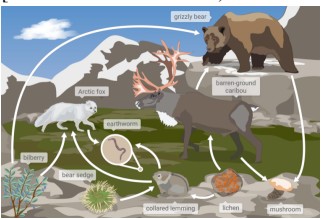

| | |
|---|---|
| User | Context: Below is a food web from a tundra ecosystem in Nunavut, a territory in Northern Canada. A food web models how the matter eaten by organisms moves through an ecosystem. The arrows in a food web represent how matter moves between organisms in an ecosystem. Which of these organisms contains matter that was once part of the lichen?
A. bilberry
B. mushroom
Answer with the option's letter from the given choices directly. |
| Sequential Train | mushroom |
| Continual LLaVA | B |
| Ground Truth | B |

[*Start a new conversation, and clear the history*]

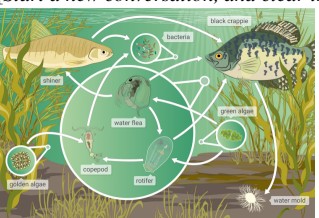

| | |
|---|---|
| User | Context: Below is a food web from Little Rock Lake, a freshwater lake ecosystem in Wisconsin. A food web models how the matter eaten by organisms moves through an ecosystem. The arrows in a food web represent how matter moves between organisms in an ecosystem. Which of the following organisms is the primary consumer in this food web?
A. copepod
B. black crappie
C. bacteria
Answer with the option's letter from the given choices directly. |
| Sequential Train | copepod |
| Continual LLaVA | A |
| Ground Truth | A |

Table 15: Comparison results between sequential training and our Continual LLaVA on the task of ScienceQA.

| **Training order:** SciQA → Text → ImgNet → GQA → Viz → REC → VQA → OCR |
| --- |
| **Evaluation results on the task of ScienceQA:** |

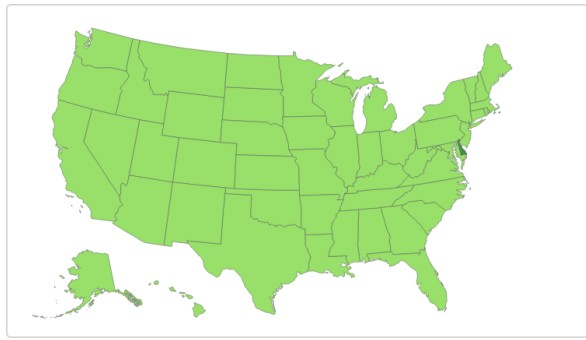

| | |
| --- | --- |
| User | What is the capital of Delaware?
A. Montpelier
B. Trenton
C. Dover
D. Georgetown
Answer with the option's letter from the given choices directly. |
| Sequential Train | delaware. |
| Continual LLaVA | C |
| Ground Truth | C |

*[Start a new conversation, and clear the history]*

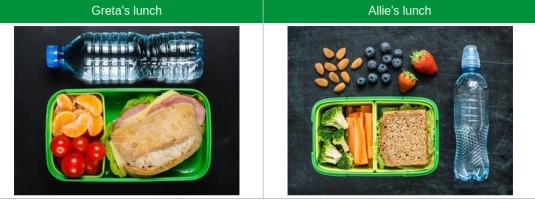

| | |
| --- | --- |
| User | Context: Trade happens when people agree to exchange goods and services. People give up something to get something else. Sometimes people barter, or directly exchange one good or service for another. Greta and Allie open their lunch boxes in the school cafeteria. Both of them could be happier with their lunches. Greta wanted broccoli in her lunch and Allie was hoping for tomatoes. Look at the images of their lunches. Then answer the question below. What can Greta and Allie trade to each get what they want?
A. Greta can trade her tomatoes for Allie's sandwich.
B. Allie can trade her broccoli for Greta's oranges.
C. Allie can trade her almonds for Greta's tomatoes.
D. Greta can trade her tomatoes for Allie's broccoli.
Answer with the option's letter from the given choices directly. |
| Sequential Train | Greta can trade to get Allie's broccoli. |
| Continual LLaVA | D |
| Ground Truth | D |

Table 16: Comparison results between sequential training and our Continual LLaVA on the task of ScienceQA.

| **Training order:** SciQA → Text → ImgNet → GQA → Viz → REC → VQA → OCR |
| --- |
| **Evaluation results on the task of ScienceQA:** |

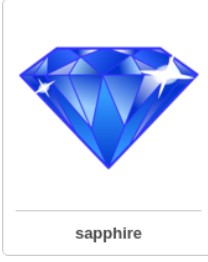 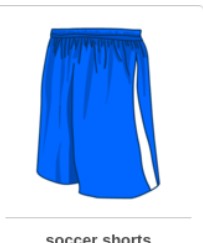

sapphire                soccer shorts

| | |
| --- | --- |
| User | Context: Select the better answer. Which property do these two objects have in common?
A. blue.
B. sticky.
Answer with the option's letter from the given choices directly. |
| Sequential Train | color. |
| Continual LLaVA | A |
| Ground Truth | A |

[*Start a new conversation, and clear the history*]

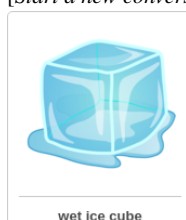 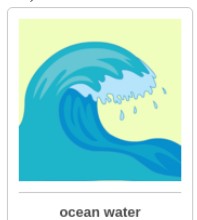 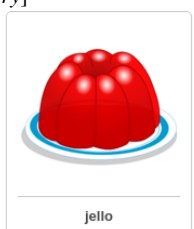 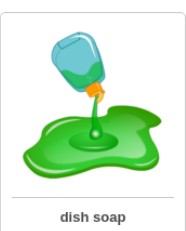

wet ice cube        ocean water         jello        dish soap

| | |
| --- | --- |
| User | Context: Select the best answer. Which property do these four objects have in common?
A. translucent.
B. flexible.
C. sweet
Answer with the option's letter from the given choices directly. |
| Sequential Train | all are foods. |
| Continual LLaVA | A |
| Ground Truth | A |

[*Start a new conversation, and clear the history*]

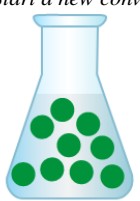 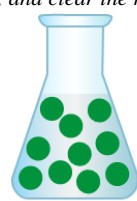

Solvent volume: 45 mL        Solvent volume: 45 mL

Solution A                Solution B

| | |
| --- | --- |
| User | Context: The diagram below is a model of two solutions. Each green ball represents one particle of solute. Which solution has a higher concentration of green particles?
A. Solution B
B. neither; their concentrations are the same
C. Solution A
Answer with the option's letter from the given choices directly. |
| Sequential Train | Solution B |
| Continual LLaVA | A |
| Ground Truth | A |

Table 17: Comparison results between sequential training and our Continual LLaVA on the task of ChartQA.

---

**Training order:** chart → document → icon → medical
**Evaluation results on the task of ChartQA:**

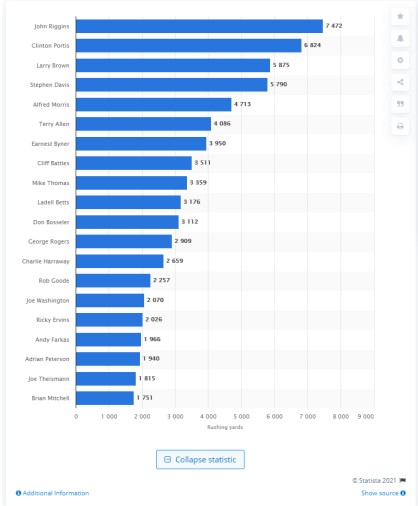

| User | What was Russia's score in the trading across borders category? |
| | Answer the question using a single number or phrase. |
| Sequential Train | 90 |
| Continual LLaVA | 99 |
| Ground Truth | 99 |

---

[*Start a new conversation, and clear the history*]

| User | Who is the career rushing leader of the Washington Football Team? |
| | Answer the question using a single number or phrase. |
| Sequential Train | ej hankins |
| Continual LLaVA | john riggins |
| Ground Truth | John Riggins |

---

Table 18: Comparison results between sequential training and our Continual LLaVA on the task of DocVQA.

---

**Training order:** chart → document → icon → medical
**Evaluation results on the task of DocVQA:**

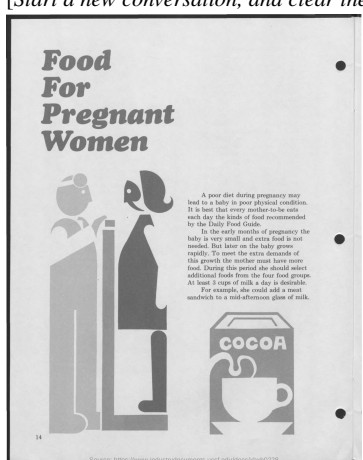

| User | Who is the person assigned monitoring responsibilities? |
| | Give the short answer directly. |
| Sequential Train | vp administrative |
| Continual LLaVA | prof. cyril o. enwonwu |
| Ground Truth | Prof. Cyril O. Enwonwu |

*[Start a new conversation, and clear the history]*

| User | How much milk is desirable a day? |
| | Give the short answer directly. |
| Sequential Train | 3 of 4 cups |
| Continual LLaVA | At least 3 cups |
| Ground Truth | At least 3 cups |

---

Table 19: Comparison results between sequential training and our Continual LLaVA on the task of IconQA.

**Training order:** `chart → document → icon → medical`
**Evaluation results on the task of IconQA:**

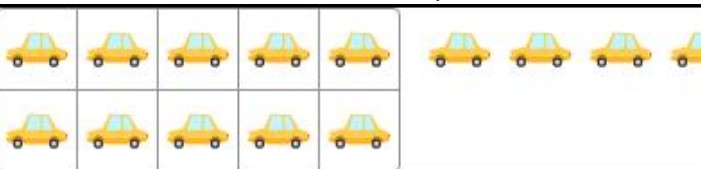

| User | Fill in the blank. Move the ruler to measure the length of the nail to the nearest inch. The nail is about (___) inches long. Give the short answer directly. |
|---|---|
| Sequential Train | 0.5 |
| Continual LLaVA | 3.0 |
| Ground Truth | 3 |

[*Start a new conversation, and clear the history*]

| User | How many cars are there? Give the short answer directly. |
|---|---|
| Sequential Train | 13 |
| Continual LLaVA | 15 |
| Ground Truth | 15 |

[*Start a new conversation, and clear the history*]

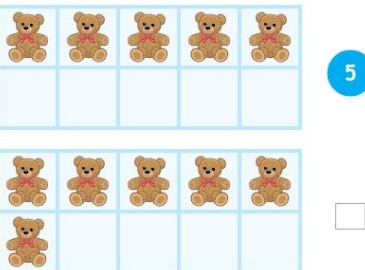

| User | There are 5 teddy bears in the top ten frame. How many teddy bears are in the bottom ten frame? Answer with the option letter from the given choices directly. |
|---|---|
| Sequential Train | 5 |
| Continual LLaVA | 6 |
| Ground Truth | 6 |

Table 20: Comparison results between sequential training and our Continual LLaVA on the task of MedicalQA.

---

**Training order:** chart → document → icon → medical
**Evaluation results on the task of MedicalQA:**

---

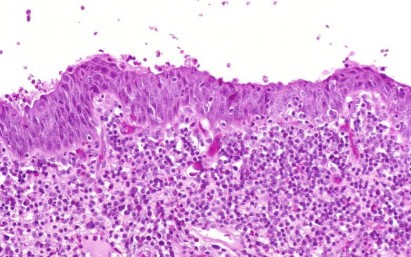

| User | How are the tumor cells? |
|---|---|
| | Answer the question using a single word or phrase. |
| Sequential Train | Small |
| Continual LLaVA | Similar to normal squamous epithelial cells |
| Ground Truth | Strikingly similar to normal squamous epithelial cells |

---

[*Start a new conversation, and clear the history*]

| User | Where is this? |
|---|---|
| | Answer the question using a single word or phrase. |
| Sequential Train | Skin |
| Continual LLaVA | Urinary |
| Ground Truth | Urinary |

---

[*Start a new conversation, and clear the history*]

| User | What does this image show? |
|---|---|
| | Answer the question using a single word or phrase. |
| Sequential Train | Gastrointestinal |
| Continual LLaVA | Squamous metaplasia |
| Ground Truth | Squamous metaplasia |

---

