# OpenReview forum: "Continual LLaVA: Continual Instruction Tuning in Large Vision-Language Models"
_ICLR.cc/2025/Conference — ICLR 2025 Conference Withdrawn Submission_

### Official Review · Reviewer_TG4q · 2024-10-29

**Soundness:** 2
**Presentation:** 2
**Contribution:** 2
**Rating:** 5
**Confidence:** 4

**Summary:**

This paper focuses on continual learning on the instruction tuning stage of large vision-language models (LVLMs). The authors propose a new benchmark, COAST, to evaluate LVLMs on three continual learning settings: 1)  domain-incremental, 2) capacity-incremental, and 3) dataset-incremental. Then, they proposes continual LLaVA, a method that uses intrinsic and contextual increment embeddings to enable continual LoRA finetuning  on LVLMs.

**Strengths:**

- Continual learning in large vision-language models is indeed an important research problem, especially when LVLMs needs to sequentially learn on different task domains. The application of continual learning on large vision-language models is somewhat novel.
- As shown in Table 1-3, the proposed method clearly outperforms baseline methods on the proposed benchmark.
- Ablation studies are provided in the paper to show the effectiveness of different components in the proposed method.

**Weaknesses:**

- In continual learning, the model should incrementally learns on **unseen novel tasks**. However, the authors start from pretrained LLaVA vision-language projectors, there might be significant overlap between the pretrained model and the continual learning task (especially detailed description). The authors do not carefully analyze the data overlap between the LVLM pretaining and the continual learning stage.
- In Figure 1, the dataset-incremental setting appears very similar to the domain-incremental setting in the proposed COAST dataset. More clarification is needed to differentiate these two.
- The method is not clear and hard to follow. In Equation 5, the intrinsic and contextual **embeddings** are added as LoRA weights. Typically, "embedding" refers to the calculated features, not the model weights. This terminology is confusing.

- The experimental sections lack important details. The authors does not mention what specific LVLM  they use in experments (LLaVA or LLaVA-1.5? 7B or 13B?). Nor do they clarify how do they initialize the finetuned parameters (e.g. the proposed embeddings).

- There is no comparison with prior work on continual instruction fine-tuning, such as [1].

  [1] Chen, Cheng, et al. "CoIN: A Benchmark of Continual Instruction tuNing for Multimodel Large Language Model." *arXiv preprint arXiv:2403.08350* (2024).

- Technical errors. e.g. L215, BPE-tokenizer only converts natural language text input into tokens (token indices), not textual embeddings.

**Questions:**

Please consider my concerns in the weaknesses section.

---

### Official Review · Reviewer_WLDy · 2024-10-30

**Soundness:** 3
**Presentation:** 2
**Contribution:** 2
**Rating:** 3
**Confidence:** 3

**Summary:**

The paper tackles the problem of continual instruction tuning, especially the visual question-answering tasks. The authors propose a benchmark, namely, COAST, which consists of domain-incremental, capability-incremental, and dataset-incremental tasks. Meanwhile, they introduce a continual LLaVA model with intrinsic and contextual LoRA weights, achieving state-of-the-art performance on the newly proposed benchmark.

**Strengths:**

+ The task of continual learning for multimodal large language models is critical in real-world applications but is under-explored.
+ The newly proposed benchmark COAST is relatively comprehensive, with three incremental dimensions: domain, capability, and dataset.
+ The proposed method achieves competitive results on COAST, surpassing baseline methods significantly.

**Weaknesses:**

1. The paper is challenging to follow, particularly in the methods section. Based on the equations and Algorithm 1, we observe that the proxy embeddings $\{k\}$ and the learnable weights $\{w\}$ are randomly initialized at the outset. This implies that the similarities between proxy embeddings and surrogate embeddings are unreliable during the initial training phase. Consequently, the top-M indices may introduce significant noise, yet only the top-M embeddings are subject to backpropagation. This raises several questions: How can the proxy embeddings be effectively learned if they are excluded from the top-M indices? How can $Z$ discard unreasonable $P$? What constitutes the ground truth for the loss function in Equation (6)? Additionally, how can we determine the ground truth for complementary task-wise correlations?

2. While it is beneficial to attempt to distinguish between the various dimensions of increment, doing so in practice proves to be challenging. For instance, a dataset-incremental approach should encompass domain-incremental and capability-incremental aspects. Consequently, such evaluation metrics may lack significant reference value.

3. Although this should be considered an early work in continual learning for multimodal large language models, there are many previous works on continual VQA, such as [A]. The paper should carefully discuss the differences between these works and the current study.
[A] Symbolic Replay: Scene Graph as Prompt for Continual Learning on VQA Task. AAAI 2023.

**Questions:**

1. The methodology section requires further clarification. The model and its learnable parameters appear challenging to train in the initial stages due to unreliable similarities.
2. How does the author address the overlapping dimensions among domain, capability, and dataset incrementality?
3. What are the differences between this work and previous continual VQA studies?

For more information, please refer to the Weaknesses section.

---

### Official Review · Reviewer_et7S · 2024-11-06

**Soundness:** 2
**Presentation:** 3
**Contribution:** 2
**Rating:** 5
**Confidence:** 3

**Summary:**

This paper considers continual learning of multimodal LLMs, where downstream tasks are sent to and learned one by one by the model. To construct the setting, the authors design the COAST benchmark that evaluates three steams of tasks under domain-incremental, capability-incremental, and dataset-incremental configurations. To solve the problem, the authors propose Continual LLaVA, a method that uses dual increment embeddings to encode task-specific characteristics and investigate the inter-dependencies across tasks. The experiments show that the method can successfully increase overall performance and alleviate catastrophic forgetting.

**Strengths:**

- The paper is generally written well.
- The proposed method closely resembles prompt-based continual learning methods, which in turn validates the effectiveness of the method.
- The experiments are well-conducted, showing that the method is better than all other baselines.

**Weaknesses:**

- While the proposed benchmark and method are good, similar ones show up in previous work [1]. In [1], the proposed method also considers LoRA selection and adaptation, but is constructed with a MoE structure. This similarity largely reduces the novelty of this paper.
- I do not see a rigorous definition of the proposed three different sub-benchmarks. For example, how do you define an increasing "capability" in capability-incremental learning? I do not see why detail description is a more increasing-capability task than conversation. Also, I do not see a clear difference between domain and dataset incremental settings, since different domains often lead to different datasets as well, and different datasets sometimes refer to different domains.
- While the experiment shows good performance, the proposed method cannot guarantee a better capability of alleviating catastrophic forgetting, since the candidate LoRA parameters are updated every task and we cannot guarantee that different tasks choose different indexes of parameters to update.




[1] CoIN: A Benchmark of Continual Instruction tuNing for Multimodel Large Language Model. NeurIPS 2024.

**Questions:**

How do you apply prompt-based method to LLaVa (baselines in the experiments)?

---

### Official Review · Reviewer_xbqG · 2024-11-06

**Soundness:** 2
**Presentation:** 3
**Contribution:** 2
**Rating:** 6
**Confidence:** 3

**Summary:**

This paper explores the continual instruction tuning of LVLMs. It first proposes a new benchmark, termed COAST, with three settings concerning domain, capability, and dataset. Then, the paper introduces Continual LLaVA, a method to conduct continual training of pre-trained LVLMs with parameter-efficient fine-tuning designs. The results on the benchmark demonstrate the effectiveness of the proposed method.

**Strengths:**

1. The investigation of continual learning in LVLMs is interesting and practical. The introduction of the benchmark is of good value and will benefit the community.

2. The design of Continual LLaVA is both efficient and effective.

3. The paper is well-written and easy to follow.

**Weaknesses:**

1. The domain of the benchmark is not sufficient enough and can be further extended, for example, multimodal scientific problems (from MMMU) math problems (from MathVerse), and autonomous driving (from DriveLM). These new domains are all important and urgent for current LVLMs.

2. The adopted LLaVA and MiniGPT-4 are old. More latest LVLMs should be experimented with to verify if the proposed method can be generalized to the most advanced base models, such as LLaVA-OneVision, Qwen-VL 2, and InternVL 2.

3. I'm curious if fine-tuning the visual encoder or adding some parameter-efficient designs into the visual encoder would help the performance

**Questions:**

See weaknesses

---

### Note · Authors · 2024-11-15

I have read and agree with the venue's withdrawal policy on behalf of myself and my co-authors.